



# A powerful lidar system capable of one-hour measurements of water vapour in the troposphere and the lower stratosphere as well as the temperature in the upper stratosphere and mesosphere

Lisa Klanner[1], Katharina Höveler[1], Dina Khordakova[2], Matthias Perfahl[1], Christian Rolf[2], Thomas Trickl[1], Hannes Vogelmann[1]

[1]Karlsruher Institut für Technologie, Institut für Meteorologien und Klimaforschung (IMK-IFU), Kreuzeckbahnstr. 19, D-82467 Garmisch-Partenkirchen, Germany
[2]Forschungszentrum Jülich, IEK-7, Wilhelm-Johnen-Straße, 52425 Jülich, Germany

*Correspondence to:* Dr. Thomas Trickl, e mail: thomas@trickl.de, Tel. 0049-8821-50283

**Abstract.** A high-power Raman lidar system has been installed at the high-altitude research station Schneefernerhaus (Garmisch-Partenkirchen, Germany) at 2675 m a.s.l., at the side of an existing wide-range differential-absorption lidar (DIAL). An industrial XeCl laser was modified for linearly polarized single-line operation at an average power of about 180 W. This high power and a 1.5-m-diameter receiver allow us to extend the operating range for water-vapour sounding to 20 km for a measurement time of just one hour, at an uncertainty level of the mixing ratio of 1 to 2 ppm. This was achieved for a vertical resolution varied between just 0.2 and 0.6 km in the stratosphere and could be improved for stronger smoothing. The lidar was successfully validated with a balloon-borne cryogenic frost-point hygrometer (CFH). In addition, temperature measurements to altitudes around 87 km were demonstrated for one hour of signal averaging. The system has been calibrated with the DIAL, the CFH and radiosondes.

*Key words:* Lidar, Raman lidar, water vapour, temperature

## 1 Introduction

Water vapour UTLS in the upper troposphere and lower stratosphere (UTLS) is the key factor controlling how much thermal infrared radiation escapes from the atmosphere into space (e.g., Kiehl and Trenberth, 1997; Schmidt et al., 2010; Lacis et al., 2013). In a warmer climate the atmosphere takes up more water vapour from the sea surface. However, this increase could be counteracted by additional cloud formation and precipitation. Also vertical exchange processes could change in a warmer climate (Trickl et al., 2010; 2020). Water vapour trends in the troposphere derived from observations are discussed in literature. Paltridge et al. (2009) report negative trends for the period 1973 to 2007 at all free-tropospheric altitudes in NCEP re-analysis data, in particular in the upper troposphere, in contrast to the expectations from climate modelling. Other studies show at least regionally positive trends (Ross and Elliott, 2001; Mieruch et al. 2008; Chen and Liu, 2016). However, they evaluate columnar quantities that are dominated by the moist boundary layer where thermal radiation is trapped by water vapour anyway. In the lower stratosphere, the Boulder series a trend reversal from positive to negative occurred around 2000 (Hurst et al., 2011), but the pronounced positive trend during the early phase since the late 1980s is not confirmed for other locations (Solomon et al., 2010; Hegglin et al., 2014). Due to its role of water vapour as the most important greenhouse gas the optimization of high-accuracy, range-resolved vertical sounding instrumentation covering the entire free troposphere and the lower stratosphere has become more and more important during the past two decades (Kämpfer et al., 2013). All the most commonly



used sensors used for routine measurements have limitations. Operational radiosondes have been greatly improved within the troposphere in recent years, but deficiencies exist in the very cold tropopause region and the lower stratosphere where the sensors exhibit slow response and low sensitivity (Miloshevich et al., 2006; Vömel et al., 2007a; Steinbrecht et al., 2008; Kämpfer et al., 2013). Balloon-borne cryogenic (CFH) sondes (Vömel et al., 2007b; 2016; Kämpfer et al., 2013; Hurst et al., 2016) and Lyman alpha hygrometers (Kley and Stone, 1978; Weinstock et al. 1990; Khattatov et al., 1994; Hintsa et al., 1999; Zöger et al., 1999; Kämpfer et al., 2013), though being highly accurate, are rarely used in dense routine measurement programmes due to their elevated costs. Ground-based microwave radiometers have an excellent temporal coverage, but their application is limited to the lower and middle troposphere (Westwater, 1978; Han and Westwater, 1995; Solheim and Godwin, 1998) and altitudes above 20 km (Nedoluha et L., 1997; Deuber al., 2004; 2005; Kämpfer et al., 2013) with somewhat limited vertical resolution. The value of satellite-borne measurements (Kämpfer et al., 2013) is limited by the considerable spatial averaging that results in a loss of information due to the high variability of water vapour even in the lower stratosphere (Zahn et al. 2014), but can yield reasonable averages and global coverage (e.g., Solomon et al., 2010).

There is just one long quantitative ground-based sounding series of stratospheric water vapour, obtained with the Boulder balloon-borne cryogenic frost-point hygrometer (Scherer et al., 2008; Hurst et al., 2011). These measurements have been carried out since 1980 at intervals of about one measurement per month. Because of the considerable variability of water vapour up to at least the UTLS more frequent measurements with good vertical resolution are desirable (Müller et al.., 2016). This variability is caused to a major extent by transport-induced patterns. Injections of water vapour into the stratosphere occur not only in the tropics (Rosenlof, 2003), where also freeze-drying has been claimed to matter (see, e.g., the discussions by Peter et al. (2003), Luo et al. (2003), Jensen et al. (2007) and Zahn et al. (2014)), but also in the jet-stream regions (Stohl et al., 2003; and references therein). Troposphere-to-stratosphere transport (TST) in vicinity of the jet streams, e.g., by overshooting moist warm conveyor belts (WCBs) can be expected to yield a significant contribution (Stohl, 2001; Zahn et al., 2014), although possibly diminished by dehydration due to cirrus–cloud formation (cirrus clouds being almost ubiquitous in WCB air probed by our lidar systems). It is reasonable to assume that water vapour transported into the lower stratosphere (LS) by TST is an important target for vertical sounding with enhanced temporal density. Also the opposite mechanism, stratosphere-to-troposphere transport (STT), is much more important than previously thought, in Central Europe after some increase over several decades (Trickl et al., 2010; 2020). Growing STT can contribute to a lowering of the tropospheric humidity.

Lidar-based measurements have the potential of good temporal and vertical resolution and are, therefore, attractive for resolving transport-related concentration changes. However, the use of lidar systems for water vapour implies a major challenge due the strong decrease of both the backscatter signal and the water-vapour concentration with altitude. Despite the problems related to the extreme signal dynamics the NDACC (Network for the Detection of Atmospheric Composition Change) lidar working group has strongly advocated to develop powerful ground-based lidar systems with UTLS capability, with focus on the Raman lidar technique. Several Raman lidar systems have already reached a reasonable UTLS performance (Congeduti et al., 1999; Whiteman et al., 2010; Dionisi et al., 2012; Leblanc et al., 2012; Dionisi et al., 2015, Vérèmes et al., 2019). Whiteman et al. (2010), Leblanc et al. (2012) and Vérèmes et al. (2019) demonstrated vertical ranges extending to more than 20 km a.s.l. for averaging over many hours.

The most important detection barrier in the lower stratosphere is the very small mixing ratio of water vapour of 4 to 5 ppm (e.g., Hurst et al., 2011). In principle, this would require a highly sensitive approach. Measurements of



molecules in a range far below one part per trillion with respect to normal conditions can be achieved in the
laboratory even under restrictive conditions (e.g., Trickl and Wanner, 1983; Trickl et al., 2010). However, a
fluorescence lidar approach cannot be used for atmospheric $H_2O$ because it electronically absorbs in the vacuum
ultraviolet spectral region and undergoes photo-dissociation as concluded from the diffuse bands (e.g., Yoshino
et al., 1997). As a consequence, lidar measurements of $H_2O$ in the lower atmosphere are restricted to the
differential absorption lidar (DIAL) and Raman scattering methods. The detection sensitivity and the range of
the DIAL method is limited by the signal noise of the absorption measurement. Raman scattering is the least
sensitive approach. However, night-time Raman scattering is a background-free method. Thus, the water-vapour
signal can, in principle, be driven to any level by enhancement of the laser power and the diameter of the
receiver, as long as allowed by financial or technical restrictions. Very importantly, a Raman lidar can be
operated at wavelengths for which absorption in the atmosphere is negligible.
For a Raman lidar calibration with an external source is an important issue: The optical transmission data of a
Raman lidar and the Raman scattering cross sections cannot be determined with sufficient accuracy. In addition,
a degradation of the components must be taken into consideration. Thus, a trace-gas Raman lidar routinely
operated over an extended period of time must be repeatedly calibrated with external references and the stability
of the calibration must be verified. Mostly, radiosonde measurements are used as reference (e.g., Leblanc and
McDermid, 2008; Dionisi et al., 2010), but also calibration with $H_2O$ column measurements are reported (Barnes
et al., 2008; Vérèmes et al., 2019). The Vaisala RS 92 radiosonde operationally used by weather services for
many years features a high accuracy level in the troposphere (Miloshevich et al., 2006; Vömel et al., 2007a;
Steinbrecht et al., 2008). However, the Raman lidar systems are not necessarily located at routine sounding
stations. Even for on-site sonde launches the sondes usually rapidly drift away from the lidar which frequently
results in discrepancies due to the high spatial variability of water vapour (Vogelmann et al., 2011; 2015).
Infrequent comparisons with sondes necessitate additional performance control such as built-in lamps (Dionisi et
al., Leblanc and McDermid, 2008; 2011; 2012; Whiteman et al., 2011) or monitoring the radiation backscattered
from air or nitrogen.
At Garmisch-Partenkirchen, we first concentrated on the differential-absorption-lidar (DIAL) technique for
measuring free-tropospheric water vapour (Vogelmann and Trickl, 2008; Trickl et al., 2013-2016; 2020). This
system has the great advantage of a good daytime performance. In recent years a high-power Raman lidar has
been built that extends the range of the DIAL into the lower stratosphere with a data-acquisition time of 1 h.
Both systems are operated side by side at the Schneefernerhaus mountain station (UFS, Umweltforschungssta-
tion Schneefernerhaus, 47° 25' 00'' N, 10° 58' 46'' E) at an altitude of 2675 m, which offers the possibility of
direct and accurate calibration of the Raman lidar. The DIAL has been thoroughly validated and is free of bias at
an uncertainty level of 1 % or less (Vogelmann, et al., 2011; Trickl et al., 2016). Both system probe the same
atmospheric volume and can be very reliably compared up to about 8 km where the DIAL data start to become
noisy.
The large system allows us to make temperature measurements up to the mesosphere based on an established
approach for inverting the Rayleigh backscatter signal for 355 nm (Hauchecorne and Chanin, 1980). In this way,
not only the primary green-house gas, but also the most important climate parameter is provided.
In this paper we report the development and the current state of the Raman lidar. We describe the steps to
achieve up to 180 W of linearly polarized and single-line output from a modified industrial xenon-chloride laser
(308 nm) (Sect. 2), and the development of the far-field receiver receiver featuring a primary mirror with a
diameter of 1.5 m (Sect. 3). Parallel to the ozone DIAL at IMK-IFU (Trickl et al., 2020) a significant step



forward in signal processing was made. The highly satisfactory lidar performance is demonstrated by examples
of 1-h atmospheric measurements, also including a temperature measurement up to 87 km (Sects. 4 and 6).
Finally, conclusions and suggestions for upgrading the lidar are made (Sect. 7).
**2 Laser System**
**2.1 General Description**
Figure 1 gives an overview of the transmitter section of the new UFS Raman lidar system in the rear part of the
lidar laboratory (see also Table 1). The transmitter consists of a high-power laser, a 2.5:1 cylindrical beam
expander to achieve a less intense 40×40 mm$^2$ quadratic beam (f = −100 mm − f = 250 mm combination,
transmittance T = 0.9985 % per lens), a hydrogen Raman shifter and a motorized (Astro System Austria, ASA)
beam steering mirror (not shown). The 0.5-m-diameter beam-steering mirror sending the radiation into the
atmosphere is located in a vertical emergency exit shaft outside the laboratory. All dielectrically coated optics, in
particular the large-diameter mirrors, were supplied by Laseroptik G.m.b.H. unless explicitly stated differently.
The efficiency of Raman scattering scales as $\lambda^{-4}$ and, thus, is the highest in the ultraviolet (UV) spectral region.
Here, the by far most powerful radiation sources are excimer lasers. The radiation source used in our system is a
big XeCl laser system with a power of 350 W (pulse energy 1 J, repetition rate 350 Hz, pulse length 80 ns) in
energy-stabilized mode of operation that is normally used for industrial applications (Coherent Göttingen
(formerly: Lambda Physik), model Lambda SX 350C, size (l×w×h) = 2.500 m × 0.850 m × 1.925 m). The very
high power of this laser system is much more important than the single-pass absorption loss in ozone at the
operating wavelength of 308 nm (Sect. 4.4).
The laser was transported to UFS by a cogwheel train of Zugspitzbahn A.G. There, it could be lifted to the 7$^{th}$
floor of the building with the large elevator of UFS and then to the 8$^{th}$ floor with two pulleys, after removing the
stairs.
As a consequence of its primarily industrial application, the laser system is operated under computer control
providing energy stabilization and numerous safety features. This is highly helpful for the planned automatic
operation of the lidar system. However, a high beam divergence of nominally 1 mrad and 4 mrad in two
perpendicular transverse orientations, random polarization and a three-line spectrum as shown in Fig. 2 are
insufficient for the requirements of the lidar. Therefore, an approach had to be found for overcoming these
disadvantages, considering the dangerous power level of this laser.
For our lidar concept a linearly polarized narrowband radiation is needed. Injection seeding with a XeCl master
oscillator with these properties was the premier choice because this could have resulted in maintaining high
average power. However, this idea was given up because of the manufacturer pointed out that there was no easy
way of synchronization because of the specified 25-μs pulse-to-pulse jitter of the big laser, and because of the
considerable additional complexity and costs.
Instead, an intra-cavity solution was chosen. The resonator was stretched as shown in Fig. 3. The intra-cavity
laser beam is first converted to an approximate squared cross section with another 2.5:1 cylindrical telescope in
order to reduce the intensity in the new rear section. It is then fed through a Brewster-angle thin-film polarizer
(transmittance 96 %) and a custom-made 70-mm-diameter Fabry-Perot etalon with 0.10 mm plate distance (SLS
Optics Ltd.; R = 54 %, T$_{min}$ ≈ 7 %, T$_{max}$ = 95.4 %) to reach the 75-mm-diameter end mirror. The large diameter
of the etalon is expected to provide strong reduction of ablation of material by scattered UV radiation and the
resulting ageing of the etalon plates. The chosen plate distance sets the free spectral range exactly to twice the
wavelength difference between the two groups of emission lines in Fig. 2. When setting the transmission





maximum to the short-wavelength component (307.955 nm; all wavelengths in this paper are specified for
vacuum) the gain at the wavelength pair around 308.2 nm is suppressed, despite the residual transmittance of
about 7 %. Just the direct first-pass forward emission estimated by the manufacturer to about 7 mJ cannot be
avoided.
The beam divergence with our long cavity was smaller than that determined by the manufacturer. We measured
a burning spot of $2.0 \times 1.2$ mm$^2$ generated on a metal plate by focussing with the $f = 2.0$-m lens in front of the
Raman shifter, corresponding to a divergence of $1.0 \times 0.6$ mrad$^2$. After the 5:1 beam expansion the beam
divergence is 0.2 mrad or less, an important prerequisite for ensuring a moderate size of the focal areas in the
very large receiver and its polychromator.
Three $43 \times 43$ mm$^2$ sand-blasted square apertures were inserted into the extended rear part of the cavity. In this
way damage of components by reflections caused by accidentally rotating the etalon beyond the needed angular
range is avoided. This can become a serious problem at a high repetition rates.
**2.2 Laser Testing**
***General Remarks***
Despite the pronounced intra-cavity losses after multiple passes through the laser cavity the maximum pulse
energy achieved at repetition rates below 100 Hz is about 0.75 J. We explain this by fresh gain generated all
along the 80 ns of laser emission and by 92 % of the amplified energy being emitted after each round trip. Thus,
the losses do not matter similarly as in a cavity with higher reflectance of the output mirror.
***Emission Spectrum***
For spectral analysis we built a 6.15-m grating spectrograph with spare $f = 250$ mm cylindrical lenses of the laser
cavity, a thin adjustable slit (OWIS) and a 1800-lines/mm grating, and a CCD camera (OPHIR) for recording the
spectrum. The spectrograph was used in second order, the third and higher orders not being detectable. The
spectral resolution was reasonable, but lower than that in Fig. 2 which was measured by Coherent in a high
grating order. During these test measurements the polarizer was not yet installed.
We first horizontally rotated the etalon with a perceivable vertical tilt angle. With this setting single-line
emission was achieved over a wide spectral range even exceeding that of the lines in Fig. 2, but with changing
pulse energy. When the etalon was oriented perpendicularly to the laser beam the full emission spectrum was
seen (lower panel of Fig. 4). We then slightly tilted the etalon vertically to the next power minimum and tuned it
just horizontally. The spectral composition changed as a function of the angle and the pulse energy could be
optimized on each of the two peaks. The upper panel of Figure 4 shows an example for maximized emission on
the short-wavelength component for a repetition rate of 300 Hz, and, after the end of this measurement, another
one for 50 Hz without etalon for spectral calibration.
The contribution of the longer-wavelength doublet for an optimum etalon angle is less than 0.5 %. This value is
in reasonable agreement with the 7 mJ of initial forward emission mentioned above, considering that almost one
half of this weak broadband emission goes into the correct wavelength component (Fig. 2). Towards higher
wavelengths a rising background (presumably from reflections or diffraction) prevents clear analyses of potential
further contributions.
For monitoring the emission spectrum an inexpensive computer-controlled miniature grating spectrograph is
used (Ocean Optics, HR 4000; $\Delta\lambda = 0.07$ nm). The performance of this spectrograph is highly satisfactory and



stable as determined from a comparison of the 308.955-nm emission that is reproducibly obtained for maximum
laser emission. Both the emission around 308 nm and 353 nm are within the limited measurement range.
In Fig. 5 we show a typical spectrum obtained with the HR 4000. The line shape is slightly asymmetrical with
higher wavelengths indicated at the top than at the bottom. The etalon angle was not fully optimized to show the
small impurity peak at 308.4 nm that is located at twice the distance between the strong line groups in Fig. 2 and
is, thus, most likely corresponds to another, weaker line of XeCl. For the highest powers achieved this impurity
grows, but stays in the range between 1.0 and 1.5 %. Further suppression would require an etalon with a slightly
larger free spectral range.
Given the specified 0.05-nm resolution of the HR 4000 spectrograph the laser bandwidth is approximately 0.05
nm. This is larger than the 0.0357 nm in the spectrum measured by Coherent in a high grating order (Fig. 2).
Therefore, we expect an emission bandwidth of less than this value.

### *Polarizer*

Linear polarization is mandatory for single-line stimulated Raman shifting (Kempfer et al., 1994) and for the
wavelength-separation strategy in our receivers (Sec. 3.2). Therefore, a thin-film polarizer was mounted in the
extended laser cavity, in the expanded section of the beam where the intensity is reduced. Despite the widened,
quadratic beam profile the substrate and the holder get rather warm after long operation of the laser at full power.
This is caused by the absorption losses due to a maximum transmittance of just 94 %. Nevertheless, the degree
of polarization of the laser output is as high as 99.4 %, in agreement with the expected 3.5 mJ (Sect. 2.1) of
forward emitted radiation with wrong polarization after the first passage though the laser medium.
Laseroptik meanwhile promised the capability of producing thin-film polarizers with more than 99 %
transmittance (as demonstrated for the polychromator). This would significantly reduce the thermal load and the
intracavity radiation losses.

### *Alignment drifts*

A careful warm-up procedure was seen as mandatory because of the long resonator. Any small thermally
induced misalignment leads to a pronounced rotation of the laser beam inside and outside the cavity which can
lead to damage of components. Horizontal misalignment of the cavity starts to progress with growing repetition
rate that requires to rotate both the etalon and the end mirror horizontally. If the optical surfaces of the etalon
stay perfectly parallel the latter is difficult to understand and is tentatively ascribed to a combination of a slight
mutual distortion of the etalon plates and the cylindrical telescope. Vertical corrections are mostly negligible.
Warm-up has been performed in 50-Hz steps. For each step, etalon and end mirror are realigned for maximum
power after about five minutes of thermal equilibration. Maximum power corresponds to optimum beam
pointing and optimum spectral purity, which is highly welcome in view of automatic control of the modified
laser. At the end a highly stable operation of the laser is achieved over many hours rarely requiring intervention.
For safety, six sand-blasted aluminium apertures were added as shown in Figs. 1 and 3. As mentioned, inside the
laser cavity even weak reflections can lead to damage at maximum repetition rate. Outside the laser head the
apertures also help to control the beam pointing.

### *Laser Pulse Energy*

In Fig. 6 the dependences of the pulse energy on repetition rate and load voltage, measured with the modified
system, is shown. For each measurement both end mirror and etalon were optimized.
The maximum pulse energy for a load voltage of 1.95 kV was 797 mJ without etalon and 765 mJ with the etalon
installed. This is much less than the 1.24 J at 1.95 kV and 300 Hz repetition rate achieved with the laser at the
factory. Of course, there are considerable intra-cavity losses. These losses are mostly caused by the polarizer and
the etalon, but perhaps also by deficiencies in imaging in the cylindrical telescope or by achieving less round
trips within the elevated-gain period due to the longer cavity. However, the overall losses are considerably
stronger than the optical losses, as we estimate from the moderate reduction in pulse energy when inserting the
etalon. We conclude that the most important drop in power is caused by the reduced number of round trips in the
extended cavity.
With growing repetition rate the energy first increases, but above 150 Hz it starts to drop considerably. This
behaviour is not similarly pronounced without the etalon as shown for comparison. It is, thus, ascribed to thermal
stress in the etalon. The optimum pulse energy at 350 Hz achieved for clean optics was 515 mJ, resulting in a
power of 180 W, one order of magnitude higher than in 355-nm Nd:YAG-based water-vapour Raman lidar
systems in the past. The power slowly decreases further during a long night-time measurement period, most
likely due to growing thermal issues. Under typical conditions we have operated the lidar in the range of 400 to
450 mJ, with aged gas even less. The pulse repetition rate was set to 300 Hz because of a time limitation in the
data-acquisition system for operation with 16000 bins.
The pulse energy at low repetition rate rises from 499 mJ at 1.55 kV to 777 mJ at 2.0 kV (lower panel of Fig 6).
**2.3 Raman Shifter and Beam Expander**
As routinely done in stratospheric ozone DIAL systems we first applied stimulated Raman shifting in high-
pressure hydrogen for generating an "off" wavelength of 353.144 nm (Sec. 3.2) as a base for ozone corrections
and a high-altitude temperature Rayleigh detection channel. We assumed that a conversion efficiency of a few
per cent are sufficient for these purposes. In this way we could fulfil two goals, to minimize the loss of pulse
energy in the fundamental wavelength for maximizing the detection sensitivity for water vapour, and to reduce
the uncertainty in the pulse-energy level at 308 nm needed for calibration of the $H_2O$ Raman detection channel.
One traditional problem with stimulated Raman shifting are losses due to the generation of high Stokes orders
and due to optical breakdown, that can, according to our experience, efficiently be accomplished even with a
long focal length of 1 m (Kempfer et al., 1994; Trickl et al., 2020). Thus, we followed the design of the
stratospheric ozone DIAL at Table Mountain (McDermid et al., 1991, and personal communication) and first
selected an f = 2.0 m focussing lens. The length of the high-pressure cell is 3.6 m.
Indeed, the measurements at low repetition rates confirmed that just the first Stokes order was generated and the
transmitted pump and Stokes energies summed up to 100 %. For 780 mJ emitted by the laser (without etalon) at
a repetition rate of 10 s$^{-1}$ 19 % conversion into the first Stokes order was measured (Fig. 6). However, the 353-
nm energy conversion efficiency at high repetition rates was much lower (Fig. 6) and just observable for slightly
misaligning the cylindrical telescope in front of the laser (Fig. 1). With perfect collimation the first-Stokes
conversion disappeared for repetition rates of roughly 100 Hz and more.
For increasing conversion efficiency, we replaced the focussing lens by an f = 1.75-m lens. This resulted in a
significantly higher conversion efficiency at a repetition rate of 10 Hz. However, the conversion broke
completely down after 0.5 min of operation when we increased the repetition rate to 100 Hz and more, which
confirms our view of overloading the hydrogen gas in the focal volume. The performance critically depends on
the alignment of the components in the laser cavity and the cylindrical beam expander outside the laser.


1. Therefore, the external beam expander was removed which resulted in a more stable performance at least up to
moderate repetition rates.

### 2.4 New Approach with a Frequency-Tripled Nd:YAG Laser

Instead of spending more time for Raman-shifting experiments, e.g., with longer focal lengths or a pair of crossed cylindrical lenses (Perrone and Picinno, 1997), we integrated in 2018 the injection-seeded Nd:YAG laser previously used in the water-vapour DIAL (Continuum, Powerlite 8020 Precision) into the system. This laser, modified for optimum beam quality for pumping a single-mode optical parametric oscillator, yields a reduced third-harmonic (355 nm) pulse energy of 160 mJ at a repetition rate of 20 Hz. This is sufficient for reasonable measurements (Sect. 6.2).

The use of this laser for providing the "off" wavelength has two advantages. Firstly, the full, stable power of the XeCl laser is available for the sounding of water-vapour. Secondly, the Nd:YAG laser is run delayed with respect to the XeCl laser. In this way interference of the 355-nm Rayleigh return in the $H_2O$ Raman channel is completely excluded.

The Powerlite laser is meanwhile operated under control of an external computer, and synchronized with the XeCl laser.

### 2.5 Conclusions for the laser system

Based on previously available laser specifications we had planned an average laser power of about 200 W, ensuring an order-of-magnitude increase with respect to frequency-tripled Nd:YAG lasers most commonly used in this field. Thus, the maximum single-line output of 180 W achieved in this project is acceptable. Also the high degree of polarization fulfils the requirements for the new lidar.

Nevertheless, the significant loss of power with respect to the free-running laser is a major disappointment. Solutions could come from injection seeding or shortening the laser cavity. We currently exclude injection seeding since this would add significant costs and complexity. Shortening means a removal of the cylindrical beam expander. This would enhance the intensity in both the etalon and the thin-film polarizer. However, as we learnt from Laseroptik, both optics can be meanwhile manufactured almost without optical loss. In this way, the thermal problems are minimized.

An important result is that for maximized output the beam pointing is extremely reproducible. Because of this property we have meanwhile started to develop automatic power optimization by horizontal rotation of both the etalon and the end mirror.

### 3 Receiver design

### 3.1 General Design Considerations

As also pointed out by Trickl et al. (2020) the receiver design of the IFU lidar systems follows a number of design principles:

(1) We use Newtonian telescopes for a less critical alignment.

(2) We separate the return in near-field and far-field channels because of the giant dynamical range of the backscatter signal (see Sec. 4.3).

(3) No optical elements or detectors are placed close to the focal points in order to avoid a modulation of the backscatter signal by the near-field scan of the focal point across inhomogeneously transmitting or detecting





1  surfaces. This prohibits the use of optical fibres because of their unknown input surface quality (apart from

2  coupling losses which mean throwing away a lot of the costly laser photons).

3  (4)  Particularly inhomogeneous surfaces (such as those of the photomultiplier tubes (PMTs) used in our

4       system) are placed in or very close to image planes (exit pupils) where the image spots and the light bundle

5       as a whole stay stable in space. This also ensures that drifts in laser pointing have no influence on the

6       position of the spot of the returning radiation on the detectors even for very long beam paths, resulting in a

7       long-term stability as long as the no part of the light bundle is cut off by a holder or an aperture.

(5)  The expensive interference filters are also placed in exit pupils to keep their diameter as small as possible.

9       The interference filters are placed in a collimated part of the radiation bundle to minimize angular spread. In

this way the near-field overlap is maximized.
(6)   All lenses with focal lengths below 0.2 m are anti-reflection coated in order to avoid angle-dependent
transmittances.
**3.2 Telescopes**
Two separate Newtonian telescopes are used with focal length f = 2 m and diameter d = 0.38 m (Intercon
Spacetec, taken from our former eye-safe aerosol lidar (Carnuth and Trickl, 1994; Trickl, 2010), and with f = 5.0
m, d = 1.50 m (Astrooptik Philipp Keller), respectively. The large focal length of the far-field telescope
necessitated to install the receiver system in a separate tower on the terrace above the lidar (Fig. 8). The tower
(Sirch and Hägele&Böhm) is covered by a 4.2-m-diameter astronomical dome with a 1.50-m slit (Baader
Planetarium) which had proved to be an adequate solution under the arctic conditions on the high mountain. The
entire structure is designed for withstanding wind speeds up to more than 300 km h$^{-1}$. The costs for the dome
limit its size, and the slit width determines the width of the large telescope. Tower and dome were transported to
the site by a big Kamov double-rotor helicopter (HELISWISS), the large mirror with a small helicopter from
Heli Tirol. The mirror was lowered to the terrace, from where it was moved into the tower under assistance of
two provisional cranes.
Although the frame of the large telescope is prepared for heating this turned out to be unnecessary because of a
powerful heating system inside the tower. The tall frame carries both the secondary mirrors and the two
polychromators without contact to the measurement compartment that is stepped on by the operators. The tower
can be entered by two doors at the terrace level and upstairs. The upper door allows us to access the
measurement compartment directly or to use the emergency exit also after a major snowfall.
**3.3 Polychromators and Wavelength Separation**
The final design of the polychromators is shown in Fig. 9. The optical table (OPTA G.m.b.H.) is in reality
oriented vertically with the left-hand side representing the top. The entrance of the radiation arriving from the
telescope is horizontal (see Fig. 8), i.e., rotated with respect to the drawing plane, as one can see from the change
in polarization vector (dot for out-of-plane to double arrow for in-plane orientation). The radiation bundle is
spatially filtered with a rectangular aperture with four adjustable blades (custom-made by OWIS) placed in the
focal plane. Due to space limitations the aperture is oriented perpendicularly to the beam axis. A slight tilt angle
would be superior because of the longitudinal walk of the "focus". This will be made possible in the future by
mounting additional inclined apertures in front of the PMTs. In this way, also the different diameters of the focal
points, caused by the different beam divergences of the two lasers, can be accounted for.


Several relay-imaging modules formed by confocally arranged f = 150 mm lenses ($f_1$) are seen (Sec. 3.1; see also
(Vogelmann and Trickl, 2008)). In the sections with parallel beams (with one exception) beam splitters and
interference filters are placed in or close to image planes of the primary mirror. Another confocal pair of $f_1$
lenses (not shown) is used to transfer the radiation from the focus of the large telescope to the first focal point in
the polychromator. The short-f lenses ($f_2$) image the principal mirror on to the photocathode of the
photomultiplier tubes (PMTs). The exact positions of the intermediate and final exit pupils can be nicely
identified with visible sky light after removing the interference filters.
The design in Fig. 9 differs from that described in (Klanner et al., 2012) that was used until 2017. The
modifications are related to the new laser concept (Sect. 2.3).
The specifications of the polychromators are listed in Table 2, including the lidar vacuum wavelengths and the
Raman shifts used. The Raman shifts in Table 2 sometimes differ from those in the lidar literature. The radiation
for the different wavelengths are separated by dichroic beam splitters and narrowband interference filters. This is
a highly demanding task considering the eight to ten orders of magnitude in signal between the Rayleigh and
Raman channels (Sec. 4.3). Figure 9 shows the principal polychromator design without the black walls
separating the detection compartments or surrounding the filters. In order to save costs, the optics of both
polychromators are equal for except for focal length $f_2$ that is chosen to achieve image diameters of the order of
5 mm for the different primary mirrors.
The optics (Table 3) were mostly purchased from Laseroptik G.m.b.H., with the exception of the narrow-band
interference filters and the steep-edge long-pass beam-splitters 5 and 6 (Materion Barr; beam splitter 6 is not
shown in Fig. 9). The width of the interference filter for water vapour (347 nm) was chosen to cover the entire
rather wide Q branch of $H_2O$ in order to avoid a temperature influence on the backscatter profiles. A broad-band
interference filter (IFB; Semrock) just transmitting the Raman return and that at 353 nm (355 nm) was added in
order to eliminate scattered light from illuminations inside the laboratory and from the buildings of the ski area.
Residual 308-nm contributions are also removed.
Some of the components have been replaced by new ones with better performance over the years, i.e., polarizing
beam splitter 1 (R > 99 % for 308 nm), and the interference filter in channel 2. The latter filter now suppresses
radiation at the pump wavelength to a level of about $2 \times 10^{-4}$. The low transmittance of the shorter-wavelength
interference filters is disappointing, but slightly exceeds that quoted. However, T = 55 to 65 % for a $\Delta\lambda = 0.1$-nm
filter has been achieved at 386.7 nm by the same manufacturer in the past (Whiteman et al., 2010).
Originally, a pair of 45º sharp-edge beam splitters was also used to separate the $H_2O$ channel from the 353-nm
channel (Klanner et al., 2012). This worked extremely well: no 308 or 353 nm Rayleigh background was
observed at 347 nm. As to 353 nm, the beam splitters reduce this spectral contribution by four decades, and the
interference filter suppresses the "out-of-band" spectral contributions by more than six decades. However, the
rise of the transmission function of these edge filters was not steep enough to minimize simultaneously signal
losses at 347 nm and contributions at 353 nm. This was acceptable during the test phase when mostly no 353-nm
emission was available at full repetition rate (see Sec. 2.2): The $H_2O$ signal was maximized by rotating the beam
splitters. It is obvious, that a slight angular misalignment could result in an undesirable change of the $H_2O$
calibration. The new design shown in Fig. 9 no longer contains these beam splitters and leads to a more robust
performance of the $H_2O$ Raman channel.
**3.4 Detectors and Discriminators**



The detector choice is based on the experience from our stationary ozone lidar system. The final development
stage took place parallel to that for the ozone DIAL and is described in more details in (Trickl et al., 2020).
Hamamatsu R7400U-03 tubes were chosen and integrated in an actively stabilized socket optimized for us in
1999 for our three-wavelength aerosol lidar by Romanski Sensors (RSV). The socket is now modified to deliver
optimized single-photon spikes without the ringing of the original PMTs that had previously enhanced the count
rate in our ozone DIAL up to about 5 km (Fig. 10). The power connection cable is shielded, but the shield is
grounded just on one side. The RSV socket generates a clean reference voltage (5 V), produced from the 15 V
supply voltage. The 5-V reference, corresponding to a PMT voltage of 1000 V, is then returned to the power
supply where it is divided to the adjustable final control voltage level (0 to 5 V) sent back to the detector. It is
important to note that this loop was necessary to clean the lidar signals to a level below $10^{-5}$ of the peak signal.
Signal-induced nonlinearities can be avoided for normal operating voltages around 800 V if one limits the signal
to roughly 100 mV or less. This level is high in comparison with traditional PMTs. In one measurement at 308
nm requiring to enhance the signal to more than 400 mV we detected deviations of the photon-counting signal
from the corresponding signals obtained at the Hohenpeißenberg station (43 km to the north) during the same
night suggesting a signal-induced contribution.
The output of a PMT is fed into an impedance-matched junction containing the discriminator (RSV). The output
for the analogue channel is slow, with single-photon pulses widened by a factor of two. The second branch is the
fast discriminator that emits −0.4-V constant amplitude pulses with a full width at half maximum of 0.6 to 1.5 ns,
depending on the photon pulse height. The discriminator level that can be chosen from −2 mV to lower voltages.
This is important for the six-dynode PMT and its rather small pulses. The pulse-height distribution for 800 V
peaks at about 10 mV (Fig. 11). We have applied discriminator levels between −4 and −5 mV.
An important issue for achieving a high sensitivity is a low level of dark counts photons, which normally
requires to cool the PMT (0.03 counts s$^{-1}$: Trickl and Wanner, 1981). With the PMTs used here and
discriminator levels of −4 mV no dark count was registered in 50-ns bins within one hour ($1\times10^6$ laser shots)
without cooling. The average external background for atmospheric measurements is clearly less than 1 count,
except for the H$_2$O channel (see Sect. 6). In the H$_2$O Raman channel a supply high voltage of up to the maximum
1000 V was used for maximizing the analogue signal that was of the order of just a few mV at a distance of 1 km
(3.7 km a.s.l.), because of the considerable dryness in the free troposphere. For the measurements typically 900
V were chosen.
**3.5 Transient Digitizers**
Following the other lidar systems developed at IFU since 1995 we purchased two 12-bit, 20 Hz transient
digitizer systems from Licel, each with six channels. Licel designed for this project and the ozone DIAL new,
ground-free input amplifiers. This latest version has led an unprecedented performance in the ozone DIAL with a
relative noise level of about $\pm1\times10^{-6}$ of the full voltage range after minor smoothing, yielding also highly
sensitive aerosol measurements at 313 nm despite the short wavelength (Trickl et al., 2020).
An exponentially decaying contribution of roughly $10^{-5}$ of the peak signal is present that scales as the signal
pulse area. This contribution differs in slope from that in the ozone DIAL, presumably because of additional
electronic components used. The artefact is more likely to be produced by combining the different units than by
the PMT itself. After introducing the discriminator for the photon-counting channel and the counter the
exponential wing increased and a slight undershoot occurred in addition. The interference could be strongly
reduced by adding an optocoupler to the trigger input of the counting system (Sect. 3.6). Some more
sophisticated impedance matching is necessary for achieving an ultimate performance. Examples for the
performance so far achieved are shown in Sect. 6.
Another limitation has resulted from the high data transfer produced by the chosen 16000 bins (120 km): The
repetition rate of the laser had to be limited to 300 Hz in order to allow for a reliable data storage.
**3.6 Photon counting**
Single-photon counting is mandatory in a lidar system with stratospheric capability. In order to benefit from the
temporal resolution of the PMTs we purchased MCS6 and one MCS6A five-channel photon counting systems
from Fast Comtec. Just two of them were used at the end since the analogue signal range for the near-field
receiver was found to be good enough to do without photon counting.
The signals are scanned for falling edges at intervals of 100 ps which means a maximum count rate of about 5
GHz for equidistant picosecond pulses. Three systems seemed to be necessary for our 12 detection channels
since one input channel of each MCS6A is used as a trigger input. However, testing the near-field receiver
showed that photon counting is not required there.
A bottle neck of this counting system is the sequential data transfer to the computer that limits the signal to
$1.8 \times 10^7$ $s^{-1}$ per 100 ns. The multi-channel scaler was, therefore, triggered with a delay of 10 to 20 μs with
respect to the laser pulse which resulted in a fully linear performance for $H_2O$. However, if an earlier beginning
of the individual measurement is desired on-board averaging becomes necessary that is not implemented in this
model. Another limiting issue had been the control program of the counting system that sometimes blocked the
start of the data acquisition. A new update of the program has led to more reliability, but has not been tested long
enough for a conclusion.
**3.7 System Control**
The electronic components of the two DIAL systems (Ingenieurbüro W. Funk) are ground-free. The trigger pulse
is derived from a photodiode and subsequently distributed into numerous output channels via optocouplers. The
supply voltages are transferred to the different devices in shielded cables. The shields of the cable leading to the
PMTs are open on the side of the detectors. The supply voltage can be set by the lidar PC via an $I^2C$ bus. Electro-
magnetic interference in the lidar signals from outside (e.g., the laser) has been kept at a negligible level by using
doubly shielded cables (Suhner, G03332; the outer shield is left open on one side) and ground-free circuits.
The data acquisition of the lidar system is controlled from a central Linux computer via a perl program and
ethernet. The Licel transient digitizers are fully read every 10 seconds. At a repetition rate of 300 Hz this allows
for an integration without overflow due to 24-bit depth for each unit. This data stream is subsequently integrated
for each channel by the controlling program until the end of the measurement after one million laser shots
corresponding to an integration time of roughly one hour. The measurement data is finally stored in an ASCII
file including meta information in the file header.
The same perl program is designed to control also the photon counting devices via ethernet communication with
the Windows based FASTComTec software. This communication does not yet work reliably for control from
outside UFS.
Meanwhile, the excimer laser can be operated via Ethernet, as well as the rotation of the etalon, the spectrometer
HR400 and a new motorized resonator end mirror. The laser power supply and cooling water pump are
controlled by Wago-SPS units (programmed in CodeSys) via a Java web interface. The beam steering mirror is
motorized and remotely controlled with a custom made software from ASA. The slit of the lidar dome, the
covers of the telescopes, the laser output mirror and the power supply of the lidar receiver are controlled with a
Wago-SPS system via a Java Web interface.
**4 Data Processing**
**4.1 Water Vapour**
A great advantage of a Raman lidar is that uncalibrated $H_2O$ concentrations are obtained in a direct way by
multiplying the backscatter signal for the full ro-vibrational Q branch by the square of distance r. Thus, small
perturbations of the signal do not matter as severely as in the DIAL algorithm that implies derivative
calculations. However, in our system the choice of a particularly powerful UV laser implicated a short operating
wavelength of 308 nm. Thus, for obtaining number densities an ozone correction must be made that is based on
the DIAL solution for the wavelengths 307.955 nm and 353.11 nm (or recently 354.22 nm).
For simplicity we have so far preferred to calculate just water-vapour volume mixing ratios. The uncalibrated
mixing ratios are calculated by dividing the $H_2O$ backscatter signal by the vibrational nitrogen Raman back-
scatter signal. Here, the influence of ozone exactly cancels because the transmitted wavelength is the same for
both Raman channels.  The photon counting data are collected at 51.2 ns per bin instead of the 50 ns in the
transient digitizers and are interpolated to match the time scale of the analogue data. In order to avoid excessive
data array sizes, we double the bin size to 100 ns during the subsequent calculations, averaging pairs of
neighbouring signals,
In the useful range for $H_2O$ up to roughly 20 km the relative noise of the nitrogen Raman signal is negligible and
no smoothing is applied. Smoothing is just applied to the Raman signal ratios that are determined separately for
the analogue and the photon-counting data. The smoothing approach is based on a numerical low-pass filtering
approach with Blackman window described and characterized in the parallel paper by Trickl et al. (2020). This
numerical filtering approach is free of ringing. The filtering interval is dynamically increased. As shown in Sec.
6 a purely quadratic dependence
$L = 1.2 \times 10^{-4}\, i^2$
as a function of 15-m bin $i$ (minimum interval size: 2 bins, $i \leq 300$) (or slightly modified for noisier data) is
adequate. In one case (5 February 2019) a third-order polynomial was used for L to achieve a better vertical
resolution in the lowermost stratosphere in the presence of a steep concentration feature. In a Raman lidar this
dependence does not require much modification from measurement to measurement, whereas in a DIAL the
strongly changing water-vapour concentration results in considerable change in absorption and, thus, of the
smoothing requirements. The definition of vertical resolution so far used by us is given by the range interval
corresponding to the 25 % to 75 % rise of the response of the smoothing filter to a Heaviside step (VDI, 1999).
For the Blackman filter the VDI vertical resolution is 19.3 % of the size of the smoothing interval. Leblanc et al.
(2016) recommend to define the vertical resolution as the full width at half maximum of a delta response which
is 34.7 % of the filtering interval for the Blackman filter. Equation 1 yields a VDI vertical resolution of 155 m at
10 km, 348 m at 15 km and 619 m at 20 km, and a delta-response vertical resolution of 277 m at 10 km, 624 m at
15 km and 1109 m at 20 km.
The role of aerosols is limited to extinction in a Raman lidar. The presence of aerosols is best judged from the
355-nm channel. The influence of extinction is very low when calculating the $H_2O$ mixing ratio from the ratio of



the $H_2O$ and $N_2$ profiles. An estimate of the extinction coefficients at the two wavelengths can be obtained from
the 355-nm data.
**4.2 Temperature**
The retrieval of temperature from lidar data is a highly demanding task. For instance, an uncertainty of 1 K
means a relative uncertainty of 0.33 % at a temperature of 300 K. Thus, a very high quality of the backscatter
signals is a prerequisite for reasonable results. For our system the two conventional methods have been selected,
evaluating the temperature dependences of the rotational Raman spectra received just below 308 nm (Arshinov
et al., 1983) and the direct retrieval of temperature from backscatter profiles (Hauchecorne and Chanin, 1980).
The retrieval of temperature profiles from rotational Raman backscattering has not yet been optimized and is,
thus, not described here. The main problem has been that the first generation of 307.390 nm interference filters
obtained from Materion Barr did not sufficiently reject the 307.955-nm contribution. In principle, this
contribution is a reasonable reference in the absence of aerosol because it is independent of temperature. Thus,
several successful temperature retrievals could be achieved for the near-field receiver (Höveler, 2015).
The evaluation of temperature profiles directly from backscatter profiles has been tested for the Rayleigh
channels at 308 nm, 353 nm, 355 nm as well as the nitrogen Raman channel (332 nm). Due to the signal losses
caused by ozone the range of the $N_2$ channel is limited. We finally decided to invert the backscatter signal for
355 nm (Sec. 2.3). The analogue and photon counting backscatter profiles are merged into a single profile,
switching at about 28 km. The resulting profile is, again, smoothed with the Blackman filter mentioned above.
Similar to water vapour the filtering interval Δ is enhanced as (approximately)
$L = 2 \times 10^{-5} \, i^2$
as a function of 15-m bin $i$.
We follow the strategy of calculating the temperature described by Shibata et al. (1986). In a first step the
density is calculated and subsequently the temperature. However, instead of the simplified density algorithm we
use a fully quantitative Klett-type approach with downward integration from the far end (Klett, 1981; 1985). The
result is calibrated to the number density $n$ and not to the backscatter coefficient:

$$n(r) = \frac{n(r_{ref}) \; r_{ref}^2 S(r_{ref})}{r^2 S(r) + 2n(r_{ref})\sigma_R \left[ 1 + \int_r^{r_{ref}} r'^2 S(r') dr' \right]} \; ,$$
      (1)

$S(r)$ being the ozone-corrected backscatter signal, $r_{ref}$ the reference distance and $\sigma_R$ the Rayleigh extinction
coefficient. We take as a first approximation a reference value calculated from NCEP (National Centers for
Environmental Prediction, http://www.ncep.noaa.gov/) data. The NCEP values are available up to a geopotential
altitude of 50 km. Beyond this, initial guesses from the U.S. Standard Atmosphere (1976) are taken, after
converting the geopotential altitudes into real ones. The results of the inversion with Eq. 1 are then compared
with radiosonde or NCEP values in a low-noise range of the backscatter profile at moderate altitudes. If the
agreement in this reference range is not sufficient, $n(r_{ref})$ is modified, and the procedure is repeated until
agreement is reached. This approach is highly robust, a change in reference value corresponding to an
approximate parallel shift of the curves. For the selection of $r_{ref}$, it is advisable to select a position for which the
signal $S(r_{ref})$ is closest to the average of adjacent data points. In this way, the subsequent correction necessitated
by the local data noise are the lowest.



The temperature is subsequently calculated from the density by applying

$$T(z) = T(z_0)\frac{n(z_0)}{n(z)} + \frac{m_{air}}{k\,n(z)}\int_{z}^{z_0} n(z')g(z')dz' \;, \tag{2}$$

with $z$ being the altitude above sea level, $m_{air}$ = 28.9644 u (U.S. Standard Atmosphere (1976); 1 u =
$1.6605390\times10^{-27}$ kg) the mass of an "average air molecule", and g the gravitational acceleration (Mohr et al.,

5    2014),

$$g(z) = g_0\left(\frac{r_E}{r_E + z}\right)^2 ,$$

with $g_0$ = 9.80665 m s$^{-1}$ and the earth radius $r_E$ = 6356766 m.
Equation 2 immediately shows that selecting $z_0$ at the upper end of the data-evaluation range means a strong
decrease with the growing density on the way downward. As a consequence, the second term in Eq. 2 clearly
dominates the temperature about 15 km downward from $z_0$. Here, the the number density retrieved in the first
step determines the temperature. Any density error critically enters the computation of the temperature. Thus, the
range of the temperature retrieval is shorter than that of the density retrieval.
**4.3 Uncertainties**
Uncertainties u of both water vapour and temperature have been approximated by the expression

$$u = \sqrt{u_0^2 + \left(u_1\frac{r^2}{r_{ref}^2}\right)^2 + \left(u_2 S(r)\right)^2} \;, \tag{3}$$

with coefficients $u_0$, $u_1$, and $u_2$ that are adjusted by comparison with reference measurements as shown in the
examples in Sect. 6. The second term in Eq. 3, quadratic in r, reflects the quadratic rise of the noise of the
unsmoothed quantities. The reference distance r$_{ref}$ is chosen at the upper end of the data evaluation range. By the
approach with Eq. 3 considerable computation efforts have been avoided.
**4.4 Simulation of the system performance**
Before finalizing the lidar design a number of simulations of the system performance were made. Figure 12
shows the results for 200 W of laser power at 308 nm, a range bin of 200 m, 10 % detection efficiency and a
measurement time of 1 h. The atmospheric data were taken from the mid-latitude summer model of the
LOWTRAN simulation program (Kneizys et al., 1988).
It is obvious that the Raman backscatter signal for stratospheric water vapour is roughly eight orders of
magnitude smaller than the Rayleigh backscatter signal for 308 nm. This imposes extreme boundary conditions
for the optical system (Sec. 3.3). The effect of the signal loss at 308 nm due to the absorption by ozone is not
very severe up to 20 km. In comparison with the most commonly used primary wavelength of 355 nm this loss is
roughly compensated by the fourth-order frequency dependence of the Raman backscatter coefficient.
**5 Calibration of the water-vapour profiles**
The calibration of the Raman lidar by the water-vapour DIAL operated in the same laboratory is a unique chance
to overcome the restrictions imposed by the sometimes extreme variability of water vapour (Vogelmann et al.,
2011; 2015). This variability is caused by a rapid sequence of atmospheric layers of strongly different origin.





The humidity varies from very high (origin in the boundary layer) to extremely low (origin in the stratosphere).
Our routine measurements since 2007 have revealed that on 84 % of our ozone measurement days stratospheric
influence could be identified in the free troposphere (Trickl et al. 2020). This leads to a particularly strong
modulation of the humidity profile.
In Fig. 13 we show the first example of a comparison between the two lidar systems on 25 April 2013. The
measurements took place under highly complex conditions in the presence of three dry layers, two of them
clearly related to stratospheric air as follows from the almost negligible humidity. 315-h backward trajectories
with the HYSPLIT model (http://ready.arl.noaa.gov/HYSPLIT.php; Draxler and Hess, 1998; Stein et al., 2015),
run here with re-analysis meteorological data, show a 5- to 7-day descent from altitudes above 9 km over
western Canada and more than 10 km above the Aleutian Islands for the layers at 4.2 km and 6.7 km,
respectively.
This was the only case in our entire test phase in which a slight 308-nm background was superimposed on the
signal. This background could be reliably removed by subtracting a very simple exponential curve. After
calibration of the data from the Raman lidar with those from the DIAL above 5.5 km reasonable agreement was
found in a major fraction of the free troposphere. However, due to using the same electronics in that early phase
the measurements were not made simultaneously. Thus, a few differences are visible.
The strong variability becomes even more obvious from comparisons with the Innsbruck (32 km to the south-
east; shown) and Munich radiosonde (100 km to the north; not shown) ascents that differ strongly and do not
show similarly dry layers despite similar courses of the trajectories calculated for these sites in comparison with
those for the lidar station. This example demonstrates that simultaneous calibration of the Raman lidar with the
quality-assured DIAL (e.g., Trickl et al., 2016) is mandatory. Unfortunately, comparisons have no longer been
possible after 2014 due to a permanent laser damage of the DIAL. The development of a new Ti:sapphire laser
system with high repetition rate is under way.
The stability of the calibration can be monitored by using the signals of the 308-nm, 332-nm and 355-nm
channels outside ranges affected by aerosol.

## 6 Measurements in the Atmosphere

After the completion of the lidar systems testing started in autumn 2012. The measurements demonstrated the
perfect suppression of interference from the other channels in the water-vapour channel by spectral filtering and
shielding. This achievement implies, according to the simulations in Sec. 4.3, a suppression of more than nine
decades of 308-nm background.
In early 2015, the near-field receiver was completed and performed well. Even rotational Raman retrievals with
a temperature noise level of 1 K were achieved (Höveler, 2015). In addition, single-photon counting successfully
entered operation for the far-field receiver, but was given up for the small telescope because of the excellent
analogue performance. In the following, we show results just for the far-field receiver since a good system
performance at high altitudes has been the main goal of this project.

### 6.1 Water Vapour Measurements up to 20 km

*1 July 2015*

The first measurement demonstrating a detection range up to 20 km was achieved on 1 July 2015. The poly-
chromator was, still, operated under testing conditions, i.e., just provisionally optically tightened against light





from the instrument panels inside the detection compartment to facilitate alignment studies. However, it turned
out that the only significant radiation leak was the wide entrance slit of the polychromator (about 40 mm × 40
mm).
Figure 14 shows the water-vapour Raman backscatter signals for this measurement, accumulated over 1 h with a
laser pulse energy of just 295 mJ (300 Hz) due to a dirty cell window. The analogue signal was corrected just
with a simple exponential correction (740 counts)*exp($-8.5 \times 10^{-5}$*i), i being the bin number, leaving a slight
residual signal undershoot is seen for distances around 12 km that is ascribed to the parallel use of analogue
detection and photon counting (Sec. 3.4).
The peak analogue signal is about 3 mV, but is rescaled here to match the counting signal. The photon-counting
background is 155 counts $h^{-1}$ $(15\ m)^{-1}$ and most likely strongly influenced by the background from the almost
full moon. The noise is much lower, about ±25 counts (standard deviation: 12.7 counts), and corresponds to an
analogue voltage of just about ±15 nV. Smoothing with a gliding 51-point arithmetic average (red curve in Fig.
14), corresponding to a VDI vertical resolution of 375 m, yields non-negative $H_2O$ signals up to r = 19.7 km
(22.4 km).
Water vapour mixing ratios were calculated just by using the analogue data for nitrogen (corrected for a very
small exponential wing) because of missing data in the corresponding counting channel during this measurement
(Fig. 15). The calibration of the mixing ratio was very difficult since there was macroscopic mutual
disagreement of the lidar and all three radiosonde profiles inspected. A few points below 7 km where the sonde
data agree were chosen as reference. The Hohenpeißenberg mixing ratio agrees best with the lidar results in the
tropopause region and is, therefore, displayed here.
The example of 1 July 2015 is special in our test phase since there was very low water vapour around 15.7 km
(about 2 ppm). The drop is verified by the Vaisala RS 92 sonde ascent at Hohenpeißenberg in the early morning,
the sonde data becoming highly uncertain at higher altitudes. HYSPLIT trajectory calculations indicated
advection of tropical air from the Caribbean Sea above the tropopause, slightly downward shifted most likely
because of a wrong model orography at the northern rim of the Alps. In the tropics freeze drying in cirrus clouds
has been suggested to lead to dehydration and, thus, low humidity (see Sect. 1). Such an inhomogeneity is a
strong motivation for lidar work that features a potential for a good time resolution. Water vapour is an excellent
tracer for troposphere-to-stratosphere transport (TST) and there is some hope that we can study some cases of
TST in the future.
***Measurements since 2018***
The measurements since 2018 were carried out with full optical insulation of the channels including the cover of
the polychromators, with narrow entrance slit and with measurements at 355 nm with the separate Powerlite
laser. In 2018 and until 6 February 2019, a total of 14 1-h measurements and several shorter tests were carried
out during nights completely without clouds. The minimum $H_2O$ mixing ratios were 4 to 6 ppm, i.e., in the range
one would expect for the stratosphere from the literature cited in the introduction.
The finally chosen size of the horizontal entrance slit was roughly 4 mm × 8 mm, slightly larger than the
minimum that is determined by the product of the large beam divergence of the enlarged laser beam and the
focal length of 5 m (receiver). The background signal in 1 h and was mostly zero in all 7.5-nm bins (rarely 1
count) except for $H_2O$. Here, typically 3 to 5 counts were registered. In one case a narrower slit was used
(roughly 2 mm × 8 mm). This led to 1 to 2 background counts, but also to an indication of a lower backscatter



signal. This would be in agreement with the large beam diameter in the focal plane of roughly 2.5 mm as
expected from the laser beam divergence and the receiver focal length of 5 m.
The reason for the background counts in the water-vapour channel could not be fully clarified. Upper-
atmosphere air-glow spectra (Broadfoot et al., 1968; Johnston et al., 1993) show several features in the
wavelength range of the in the lidar return for $\lambda \geq 332$ nm. However, some spectral overlap also exists with the
components at 332 nm, 353 nm and 353 nm. No clarification has been possible.
*19 July 2018*
During the early hours of 19 July 2018, two subsequent measurements were made that could be compared. The
average laser pulse energy was just 380 mJ (300 Hz). The background count rate was 5-8 counts $h^{-1}$ $bin^{-1}$ for a
slightly larger entrance slit.
The mixing ratios obtained are shown in Fig. 16. The calibration of the first measurement was estimated from
the Munich sonde data for the launch at 1:00 CET. The profile for the second measurement looks completely
different which, again, demonstrates the strong atmospheric variability of water vapour. Here, the calibration of
the lidar mixing ratios was based on the Innsbruck sonde (nominal daily launch: 4:00 CET). We assume that the
horizontal homogeneity is much better in the the tropopause region, where we, thus, centred the calibration.
However, the agreement is also reasonable around 6.5 km (5 to 10 %).
The two profiles for the lidar agree quite well up to about 18 km (Fig. 17), despite the elevated signal
background. The second measurement was noisier which is reflected by the larger error bars.
It is interesting to note that the sonde data do not look as reliable in the lower stratosphere as in the July 2015
case. We speculate that this is due to a change in sonde type from RS 92 to RS 41. We have found that the RS 92
data highly realistic in our tropospheric studies in comparison with our DIAL (Trickl et al., 2014-2016). For
2018, the new sonde type exhibited a positive bias of 2-3 % relative humidity (RH) in intrusion layers.
*5 February 2019: system validation*
On 5 and 5 February 2019 several balloons with cryogenic frostpoint hygrometers (CFH; Vömel et al., 2007;
2016), standard Vaisala RS-41 SGP radiosondes (Vaisala et al., 2019), ECC ozone sondes (Smit et al., 2007) and
COBALD backscatter sondes (Brabec, 2011) were launched in the valley at IMK-IFU (9 km to the north-east of
UFS) by a team of the Forschungszentrum Jülich. The data were transmitted to a ground station installed for this
campaign at the Zugspitze summit. The combined balloon payload is well tested and regularly also used by the
GCOS Reference Upper Air Network (GRUAN) (e.g., Dirksen et al., 2014).
The CFH has an uncertainty of about 2-3 % in the troposphere and less than 10 % in the lower stratosphere.
Thus, the CFH is especially suitable for measuring water vapour under the dry conditions at the tropopause and
in the stratosphere up to altitudes of 28 km.
The first night of the campaign was clearer and these results are presented in the following. The conditions for
the comparison were excellent: the sondes rose almost vertically up to 8.5 km and then slowly drifted to the
south-east (Innsbruck). The balloons stayed within 20 km distance up to the tropopause (12.8 km a.s.l.) and
remained within 30 km from IMK-IFU up to 20 km a.s.l.
The launch times of the balloons were 18:03 CET (ascent to 16.147 km), 19:03 CET (29.475 km), and 23:00
CET (29.469 km). The profiles of the CFH $H_2O$ mixing ratio during that period mutually agreed to within 0.5
ppm between 13.0 km and 17.5 km and slightly more up to 26 km. Just two of the three lidar measurements at
UFS cover the full standard measurement time of one hour and are presented here.



The $H_2O$ Raman backscatter profile for the measurement before midnight is shown in Fig. 18. Due to a narrow
slit the $H_2O$ raw data exhibit a background of just 2.33 counts (subtracted here) with a standard deviation of 1.55
counts. Two curves with gliding arithmetic means over ±25 and ±75 bins are included that suggest a useful range
up to r = 17 km (h = 19.7 km a.s.l.). The remarkably low sensitivity limit for the averaged curve corresponds to
roughly 0.1 nV of analogue voltage. The dynamic range within the dry free troposphere and the lower
stratosphere covers astonishing seven decades.
The nitrogen Raman backscatter signal is considerably larger. Thus, the onset saturation effects can be seen in
the photon-counting data below r = 4 km. Here, the analogue data are, still, valid for at least two more downward
kilometres. The analogue signal starts to deviate from the photon-counting signal due to an exponential decay in
the signal processing mentioned in Sect. 3.5. We do not correct this effect because the photon-counting method
is used at high altitudes.
Figures 19 and 20 show the water vapour mixing ratios obtained for two measurement periods on 5 February
with 1-h lidar measurements together with those from the almost simultaneous CFH ascents. In addition, the
values for the Munich radiosonde (6 Feb, launched at 1:00 CET) are included for comparison. The grey curve
corresponds to the VDI vertical resolution used for the numerical filtering that is about 0.2 km at 14 km and 0.47
km at 20 km. Due to the moderate smoothing around 13 km the downward humidity step at 12.8 km is just
slightly widened with respect to the CFH sensor. We reduced the vertical resolution of the first measurement
around this step by introducing a third-order dependence (polynomial) of the smoothing interval (Eq. 1), but
could not improve the steepness of this step. We conclude that the width of the step in the lidar result is primarily
determined by the long data acquisition over 1 h.
The lidar was calibrated in the upper troposphere above 7.7 km yielding an almost perfect agreement with the
CFH measurements in this range. Between 7.7 km and 5.7 km it is, still, satisfactory with deviation of 5 to 10 %.
Below this altitude the agreement for the first profile was also acceptable, the lidar value lying in the middle of
the CFH mixing ratios for ascent and descent (the latter not shown for clearness). This was quite different for the
second profile recorded before midnight when the atmosphere was obviously highly inhomogeneous in space,
even on a horizontal scale of 10 km given by the almost vertical rise of the balloon. The presence of several very
thin dry layers, also over Munich, indicate a pronounced filamentation.
Below this zone the agreement is good for both measurements. This indicates a good cancellation of the overlap
functions of the nitrogen and water-vapour channels, similar to DIAL systems.
**6.2 Temperature Measurements**
*Rotational Method*
A few measurements based on the rotational temperature method were evaluated for the near-field receiver
(Höveler, 2015). The Cabannes influence was corrected for. A good performance with a temperature noise of
less than 1 K in a range up to 8 km in the free troposphere was achieved. With recently purchased new narrow-
band interference filters (Materion Barr rejection of Cabannes radiation to $2 \times 10^{-4}$) and a better polarizing beam
splitter (Laseroptik, T > 99 %) we expect a much better rejection of the Cabannes radiation.
*Rayleigh Method*
Temperature profiles based on the Rayleigh approach have been made for the wavelengths 308 nm, 332 nm, 353
nm and, finally, 355 nm. For 308 nm and 332 nm the signals must be corrected for the absorption of the radiation
in ozone. The range for 332 nm ends far below the mesosphere and is, therefore, no longer considered. For 308



nm a temperature retrieval up to 55 km was achieved. However, the backscatter signal was attenuated with a
neutral-density filter by a factor of one thousand in order to avoid detector overload. This means that, without
attenuation, a high-speed chopper must be added to cut off the signal returning from the first ten kilometres.
Then, the performance could be excellent. The 353-nm channel was successfully tested at low repetition rates
(yielding reasonable temperatures up to 52 km), but was given up because of the loss of Raman conversion at
full power.
Here, we present the first demonstration of a measurement with the separate frequency-tripled Nd:YAG laser
(Sect. 2.4) up to the mesosphere on 16 November 2018. Figure 21 shows the backscatter signals for a 1-h
measurement together with simulations for the U.S. Standard Atmosphere and a combination of the 1:00 CET
Munich radiosonde and the 13:00 CET NCEP data for our station downloaded from the NDACC web site. The
analogue signal exhibits a considerable distortion at high altitudes which we ascribe of the known (Trickl, 2010)
magnetic interference of the Nd:YAG laser. Again, a correction is not necessary because the photon-counting
data are used at high altitudes.
The strong near-field signal peak was suppressed by using a narrow aperture and by rotating the laser beam away
from the telescope axis. However, this resulted in reduced overlap as far as almost 20 km, as can be seen in the
temperature data (Fig. 22). The combined raw data were smoothed with a VDI vertical resolution scaling as
shown in Fig. 21, the maximum value staying below 2 km.
The temperature data were initialized at 87 km a.s.l. (density: at 95 km) by using the temperature of the U.S.
Standard atmosphere as the start value. The performance is surprisingly good, despite the strongly growing noise
of the raw data in this altitude range. The agreement with the temperatures from the Munich radiosonde and
NCEP is very good up to the upper end of the NCEP table (50 km) downloaded from the NDACC web site. For
higher altitudes we first compared our results with the MSIS model output calculated for our site, as
recommended by Wing et al. (2018). There is a strong discrepancy that could not be reduced by selecting the
MSIS temperature at 87 km as the start value of the retrieval: The temperature converged to the curve for the
standard atmosphere within just 15 km.
A comparison with the temperature of the Microwave Limb Sounder (MLS) during the early hours of 16
November for a position 3.5º farther to the east. Considering the difference in position the agreement with the
MLS temperature profile is quite good, with a strong similarity in structure. The temperature peak at 65 km is
present, but slightly downward shifted.
In summary we are highly satisfied by this first result. In principle, due to the very small average background
signal, Poisson effects in the photon statistics must be taken into consideration. More advanced approaches are
needed, such as that presented by Sica and Haefele (2016).
**7 Discussion and Conclusions**
The primary goals of the system development described in this paper have been to reduce the measurement time
for lower-stratospheric water vapour up to at least 20 km to one hour and to achieve temperature measurements
up to more than 80 km. These goals have been met, with a satisfactory performance. Nevertheless, a comparison
with the simulations in Sect. 4.4 clearly shows that the measured lidar signal for water vapour is considerably
smaller than predicted.
At 15 km our measurements typically yield $H_2O$ Raman returns of 2 counts per 7.5-m bin and hour. This is
converted to 53 counts for the 200-m bins used in the calculation, one sixth of the 315 counts simulated.
Roughly a factor of two is due to the lower laser paper in comparison with the 200 W assumed in the simulation.





For the rest, apart from uncertainties in the parameters used in the numerical estimate, we found that the most
likely reason for this discrepancy is that the Raman cross section used in the calculations is presumably given for
the sum of all three ro-vibrational branches. Indeed, the peak signal increased by roughly a factor of three when
we removed the 347-nm interference filter, which includes the missing attenuation by the interference filter (T =

5   0.72).

As a consequence, we carried out measurements without interference filter. However, this resulted in a much
higher stratospheric mixing ratio of 120 ppm due to insufficient blocking of 308- or 332-nm radiation. Thus, for
collecting the signal from the entire ro-vibrational band at least a broad-band interference filter (bandwidth 20
nm) must be added to reject residual contributions from the other channels and to reduce the observed 3 to 4
background counts.
The background is dominated by the size of the entrance slit. The optimum slit width is different for both lasers,
given different beam divergences. Therefore, in the future two additional slits will be used in focal points of the
347-nm and 355-nm channels (Fig. 9). This slits are easier accessible than the entrance slit which facilitates to
optimize their position and size..
There are obvious possibilities to enhance the laser power. Better transmitting intracavity optics should be
installed for significantly reducing the thermal load. A higher transmittance would, therefore, also allow us to
remove the cylindrical telescope (Fig. 3) that was introduced to reduce the intracavity intensity on the optical
components added by us. As a consequence, the resonator would become shorter and the number of cavity round
trips within the fluorescence time of XeCl would grow. As pointed out in Sect. 2.1 the reduction of the numbers
of round trips is likely to be the dominant loss factor in the extended resonator.
The calibration of the water-vapour channel was confirmed to be a key issue for the long-term operation of the
lidar. We hope that the UFS DIAL can soon be re-activated for filling this gap. Additional control by inspecting
the data from surrounding radiosonde stations or the signal level at 308 nm and 332 nm are other important tools
to ensure long-term stability of the system.
The temperature measurements with a separate, frequency-tripled Nd:YAG laser were quite successful.
Improvements could result from using a diode-pumped Nd:YAG laser with 300 or 350 Hz repetition rate,
matching that of the XeCl laser. Such lasers are meanwhile available. We expect lower pulse energies for such a
laser at 355 nm, but the currently available 160 mJ yielded too much backscatter signal anyway.
The remaining tasks will concentrate on intensifying the measurements. Lidar measurements at high temporal
resolution may yield important hints on the role of atmospheric transport, in particular TST,  on the water vapour
concentration in the UTLS. Finally, given the current debate on the climate development, an important
contribution to the question about the $H_2O$ feedback could be given.
**5 Data availability**
Data can be obtained on request from several authors of this paper (christian.rolf@fzj.de; thomas@trickl.de,
hannes.vogelmann@kit.de).
**6 Author statement**
All authors from Garmisch-Partenkirchen were involved in system development and lidar testing. The Jülich
team launched balloons at IMK-IFU.
**7 Competing interests**





The authors declare that they have no conflict of interest.
**Acknowledgements**
The authors thank Hans Peter Schmid for his interest and support. They are indebted to Werner Funk, Bernd
Mielke, Heinz Josef Romanski and Bernhard Stein for numerous important discussions and technical
improvements. Stuart McDermid sent valuable information on the feasibility of Raman lidar measurements of
$H_2O$ in the stratosphere which encouraged us to start this project. We strongly appreciate the intense discussions
with our NDACC colleagues. Wolfgang Steinbrecht generously made available results from the nearby
Hohenpeißenberg observatory, Gerald Nedoluha provided the MLS reference data. The good co-operation with
the Coherent team in Göttingen was crucial for the laser upgrading. Also crucial has been the excellent help by
Werner Moorhoff and Laseroptik who made numerous attempts for optimizing the performance of their
dielectric coatings. In particular, their tenacity in optimizing the reflectance of the large mirrors in the transmitter
at their own costs this project prevented serious damage to this project. This work has been funded by the
Bavarian State Ministry of Environment and Consumer Protection under contracts 45001226 (KIT), TLK01U-
49581 and VAO-II TPI/01. KIT acknowledges support of lidar measurements by the European Space Agency
(ESA) under Contract 4000123691/18/NL/NF (FIRMOS validation campaign). Balloon profiles utilized in this
paper have been provided within the same ESA project by the Forschungszentrum Jülich via subcontract with
KIT. The balloon activities were also partly supported by the Helmholtz Association in the framework of
MOSES (Modular Observation Solutions for Earth Systems).
The service charges for this open-access publication have been covered by a Research Centre of the Helmholtz
Association.

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



1    **Table 1. Transmitter Details**

| 2 | Laser source | XeCl laser (Coherent, model Lambda SX) |
|---|---|---|
| 3 | Laser wavelength | 307.955 nm |
| 4 | Maximum power (at 2.0 kV) | 420 W |
| 5 | Stabilized power (all lines) | 350 W |
| 6 | Single-line power | 180 W |
| 7 | Line width | 0.036 nm |
| 8 | Optimum spectral purity | 99.5 % |
| 9 | Linear polarization | 99.6 % |
| 10 | Pulse repetition rate | $350 \ s^{-1}$ |
| 11 | Raman shifted wavelength | 353.144 nm |
| 12 | Maximum Raman conversion effi- | |
| 13 | ciency (f = 2.0 m, 40 bar, $350 \ s^{-1}$) | 5 % with distorted alignment, otherwise 0 % |
| 14 | Second laser (starting 2018) | Nd:YAG (Continuum, model PL8020 Precision) |
| 15 | Wavelength | 354.8123 nm (injection-seeded) |
| 16 | Pulse energy | 160 mJ at 20 Hz repetition rate |
| 17 | Final beam expansion (f = 1.75 m) | 5.7:1 |
| 18 | Final beam dimensions | $0.20 \times 0.20 \ m^2$ |
| 19 | Final beam divergence | $\leq 0.5$ mrad |
| 20 | | |
| 21 | | |



**Table 2: Receiver Details**

| | |
|---|---|
| Primary mirrors | 0.13 m diameter, f = 0.72 m |
| | 0.50 m diameter, f = 2.0 m |
| Field of view | large telescope: about $0.8 \times 0.2$ mrad$^2$ |
| Detection wavelengths: | 306.791 nm, 307.390 nm, 307.355 nm, 331.751 nm, 346.978 nm, |
| | 353.144 nm, 354.812336 nm[a] |
| Raman shifts: | Vibrational Q branch of $H_2O$[b]: 3652 cm$^{-1}$ (centre of the stronger lines. Range |
| | of Q-branch: 3628 to 3658 cm$^{-1}$ ($\Delta\lambda = 0.36$-nm) |
| | nitrogen, $Q_6$ line (population peak)[c]: 2329.1821 cm$^{-1}$ |
| | $N_2$ and $O_2$ rotational shifts: taken from references in footnotes c and d |
| | hydrogen, $Q_1$ line[e]: 4155.2521 cm$^{-1}$ |
| Wavelength separation | polarization-sensitive beam splitters and interference filters |
| | ($\Delta\lambda = 0.75$ nm f.w.h.m. for $H_2O$, 0.25 nm otherwise) |
| PMTs | Hamamatsu R7400U-03, modified by RSV |
| Pre-amplifiers | Analog Modules, gain 1−10, bandwidth 4 MHz, sometimes used for $H_2O$ |
| Transient digitizers | Licel, 6 units, 12 bit, 20 MHz, ground-free input stages |
| Photon counting | FAST ComTec, 100 ps time bins, 7.5-m detection bins |

(a) Measured during the project described by Vogelmann and Trickl (2008)

(b) Avila et al., 2004

(c) Trickl et al., 1993; 1995

(d) Rouillé, 1992; Golubiatnikov and Krupnov, 2004

(e) Bragg et al., 1982; Dickensen et al., 2013



1   **Table 3: Specifications of the Polychromator Optics**

2   **Broadband Optics**

| Component | Diameter | Focal Length or Wavelength | Comments |
|---|---|---|---|
| $f_1$ lenses | 75 mm | 150 mm | |
| $f_2$ lenses | ** mm | 17 mm | large telescope |
| | ...** mm | 30 mm | small telescope |
| 45º-high reflectors | 75 mm | all wavelengths | S and P polarization |
| 45º-beam splitter 1 | 75 mm | 308 nm | T = 99 % P (308 nm), |
| | | | T = 99-100 % P (> 325 nm) |
| | | | T = 94 % S (355 nm) |
| | | | R ≈ 99.8 % S (308 nm) |
| 45º-beam splitter 2 | 75 mm | 308 nm | T = 63 % S, R = 37 % S |
| 45º-beam splitter 3 | 75 mm | 308 nm | R = 100 % P (308 nm) |
| | | | T = 83 % P (332 nm) |
| | | | T ≈ 90 % P (347, 355 nm) |
| 45º-beam splitter 4 | 75 mm | 355 nm | T = 99 % P (332, 347 nm |
| | | | R ≈ 99.8 % S (355 nm) |
| 45º-beam splitter 5 | 75 mm | 332 nm | R = 99.8 % P, |
| | | | T > 99 % P (347, 355 nm) |
| 45º-beam splitter 6[*] | 75 mm | 347 nm | R = 84 % P |
| | | | T = 99.3 % P (353 nm) |

23   [*] No longer used since 2018, not shown in Fig. 9

25   **Interference Filters**

| Wavelength [nm] | Bandwidth [nm] | Maximum T (large telescope) | Maximum T (small telescope) | Producer |
|---|---|---|---|---|
| 306.791 | 0.25 | 25 % | 25 % | Materion Barr |
| 307.390 | 0.15 | 27 % | 25 % | Materion Barr |
| 307.955 | 0.25 | 35 % | 32 % | Materion Barr |
| 331.751 | .....0.25 | 52 % | 43 % | Materion Barr |
| 346.978 | 0.75 | 74 % | 62 % | Materion Barr |
| 353.144 | 0.25 | 43 % | 34 % | Materion Barr |
| 354.812 | < 1.2 | > 80 % | | Alluxa |

35   All diameters 50 mm





**Figures:**

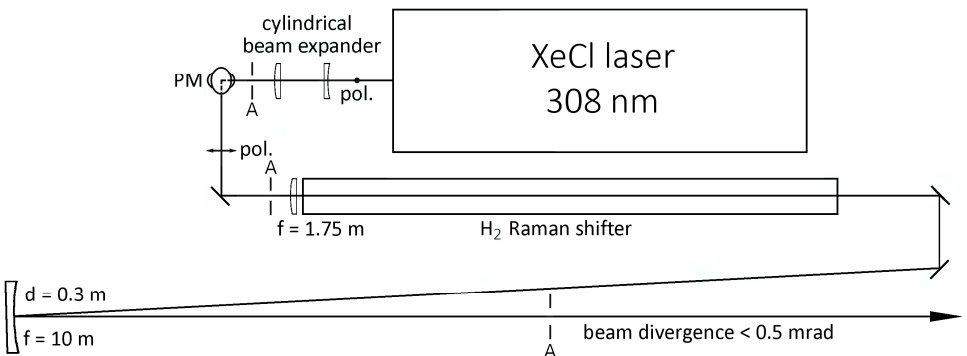

**Fig. 1.** Overview of the transmitter part of the UFS Raman lidar: The laser beam profile is expanded to a 36×36-
mm$^2$ square shape by a f = −100 mm − f = 250 mm pair of cylindrical lenses (recently removed), sent down by a
combination of two plane mirrors (rotating the polarization by 90º) before it is focussed into a high-pressure
Raman shifter 3.6 m long with a f = 1.75 m lens (initially f = 2.0 m). The beam diverges from the focal point is
made parallel by an f = 10 m concave mirror (about 180 × 180 mm$^2$) and reaches the motorized beam-steering
mirror in a vertical exit shaft outside the laboratory (not shown). Three apertures (A; 40-mm slit, 40×40-mm$^2$
square and (w = 200)×(h = 120)-mm$^2$) made of sand-blasted anodized aluminium allow to control the beam
pointing that can change during warm-up.

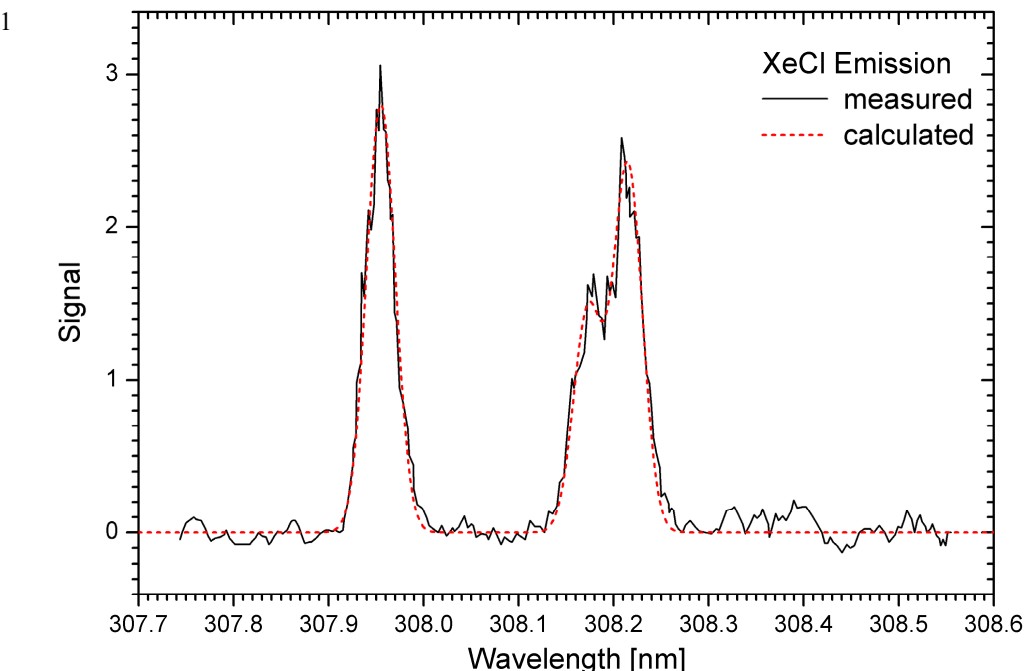

2 **Fig. 2.** Emission spectrum of a Coherent high-power XeCl laser in broadband operation (source: Coherent); the

3 dashed red curve is the sum of three Gaussian lines with centres at 308.955 nm, 308.173 nm and 308.215 nm and

4 full width at half maximum of 0.0357 nm.



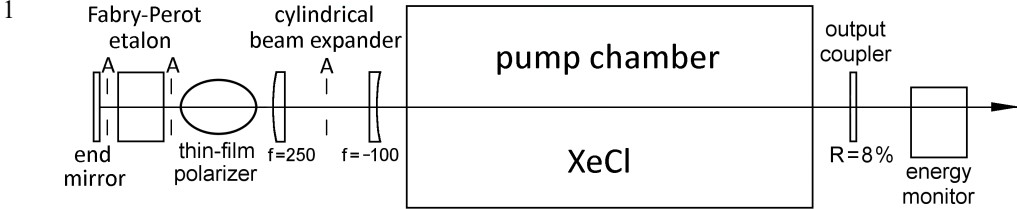

2   **Fig. 3.** Top view of the modified Lambda SX laser system; 36×36-mm$^2$ square apertures (A) are used for

3   protecting optical components from potential powerful reflections from accidentally rotated components. The

4   polarizer is oriented out of plane at Brewster´s angle.

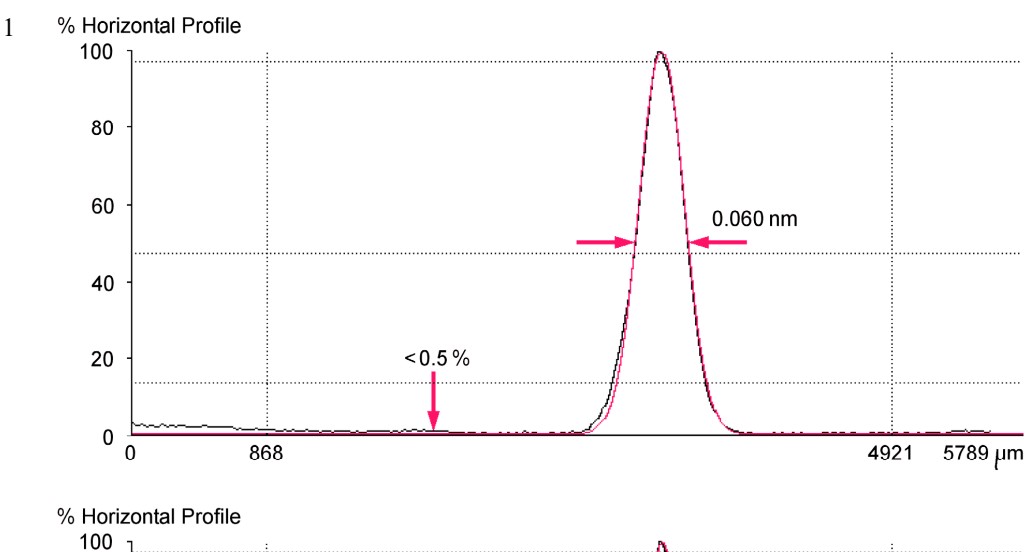

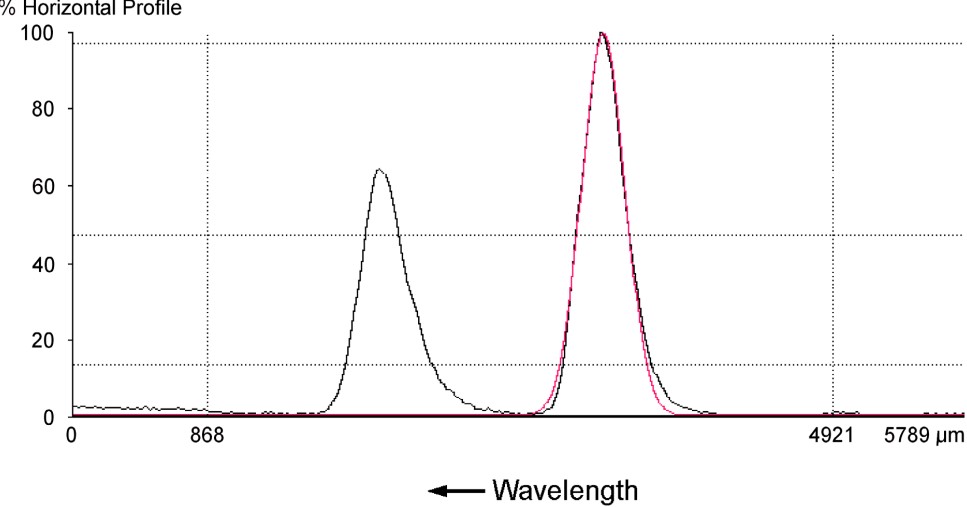

⟵ Wavelength

**Fig. 4.** Uncalibrated spectra of the laser emission with the laser running at 300 Hz (top) and 50 Hz (bottom, etalon removed); the red curves are Gaussian fits by the camera software. Asymmetries and the growing background towards higher wavelengths are to some extent ascribed to imperfections of the home-made spectrograph. The grid of the lower spectrum was slightly shifted to achieve position matching of the large peak (minor re-alignment of the spectrograph between the measurements)





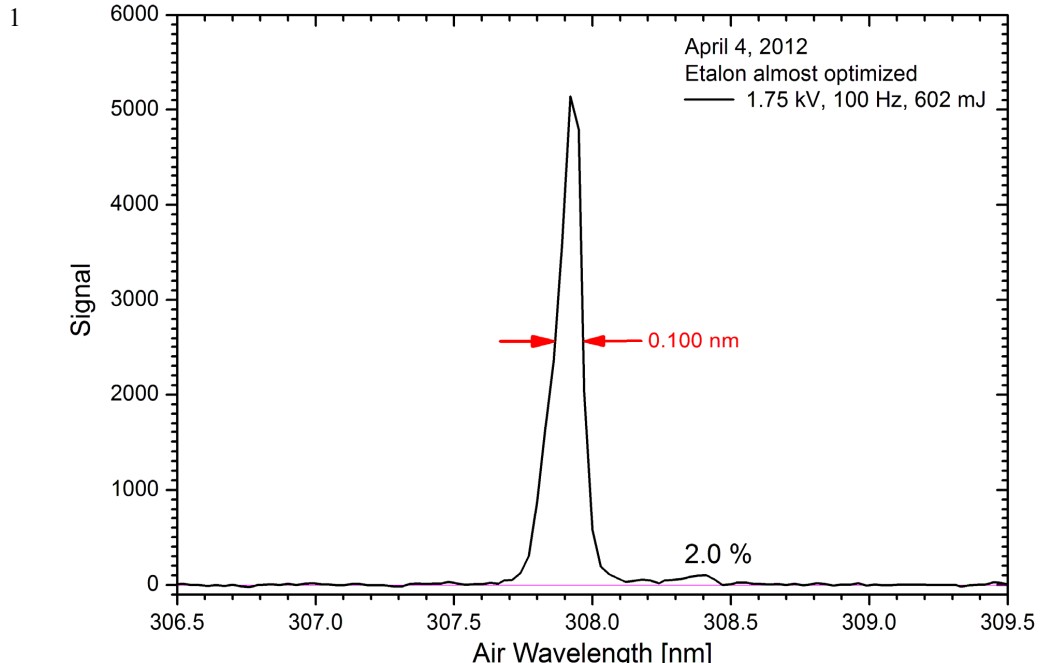

2 **Fig. 5.** Spectrum of the laser emission with almost optimized etalon angle; the laser was operated with 10 Hz

3 repetition rate, 1.75 kV load voltage and 663 mJ (including the polarizer).

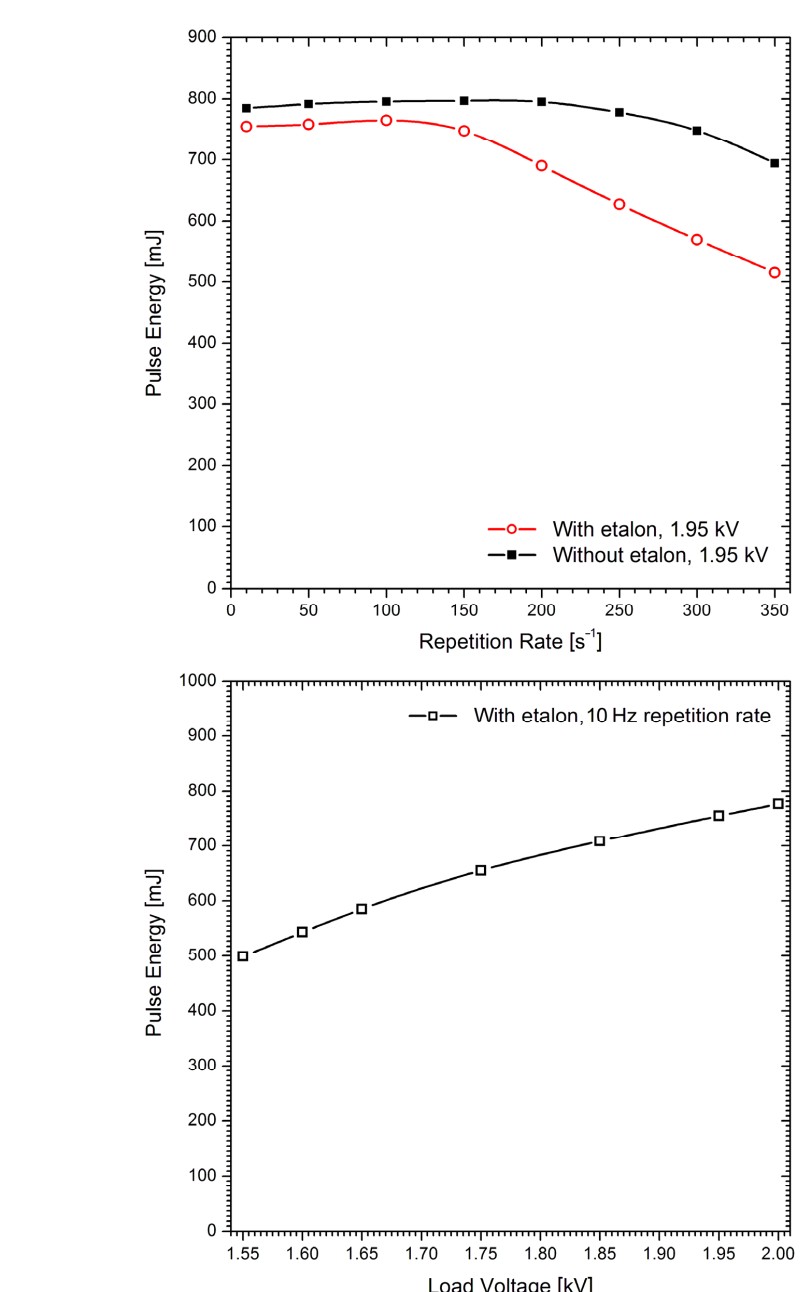

2 **Fig. 6.** Optimized pulse energy as a function of the repetition rate (top) and load voltage (bottom); for

3 comparison: The maximum pulse energy of the broadband laser as delivered is 1.25 J (at 2.0 kV and 300 Hz).



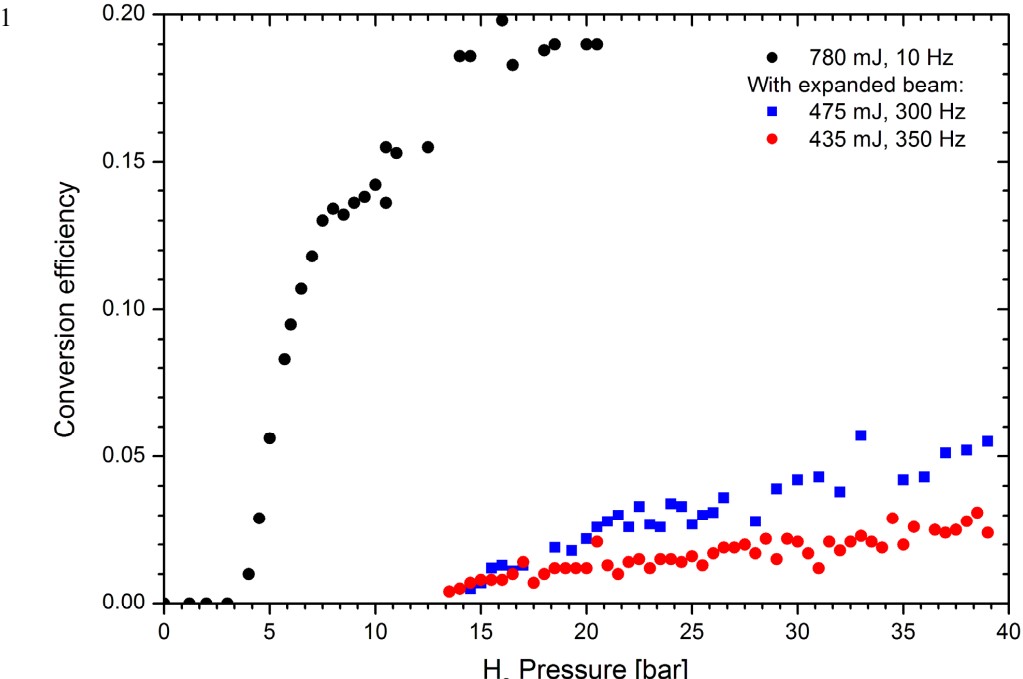

2 **Fig. 7.** Raman conversion efficiencies in hydrogen obtained for three different repetition rates (f = 2.0 m): The

3 measurements at 10 s⁻¹ repetition rate were made without etalon in the laser cavity, those at high repetition rates

4 with all components installed.



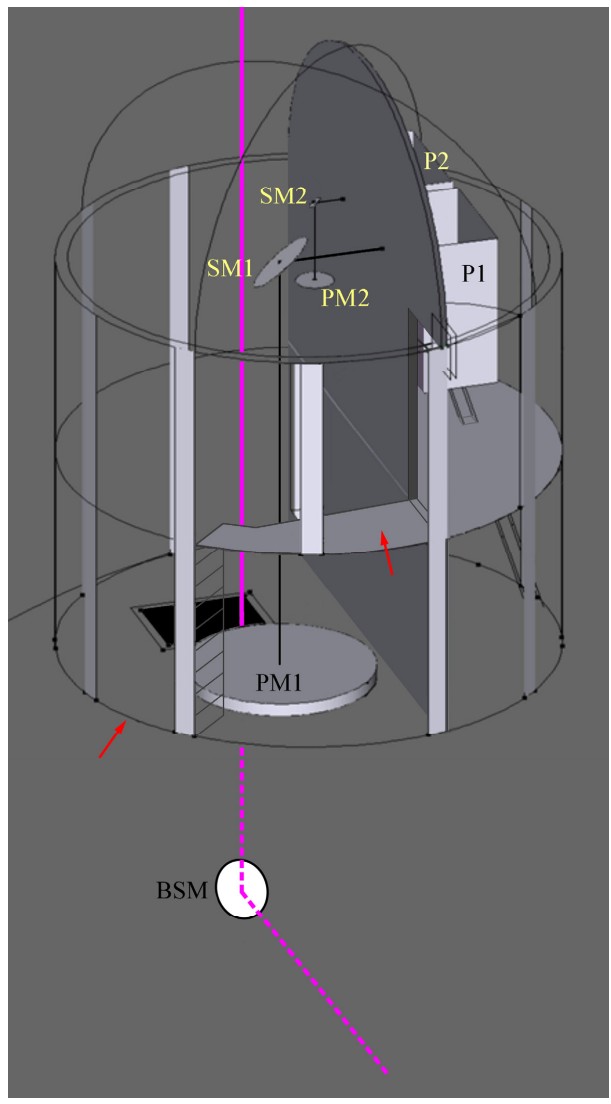

**Fig. 8.** Receiver tower mounted on the terrace above the lidar laboratory: The tower is covered by a 4.2-m-
diameter astronomical dome with a 1.5-m slit: The laser beam (violet) emerges from a former emergency shaft.
The plane formed by the axes of the large telescope and the laser beam contains the section of the laser beam in
the lower floor. This plane is perpendicular to the plane formed by the axes of the small telescope and the laser
beam. Abbreviations:

| | |
|---|---|
| BSM | beam-steering mirror |
| PM: | primary mirror |
| SM: | secondary mirror |
| P: | Polychromator |
| 1, 2: | belonging to far-field receiver, near-field-receiver, respectively |

The two red arrows indicate the two entrances of the tower.



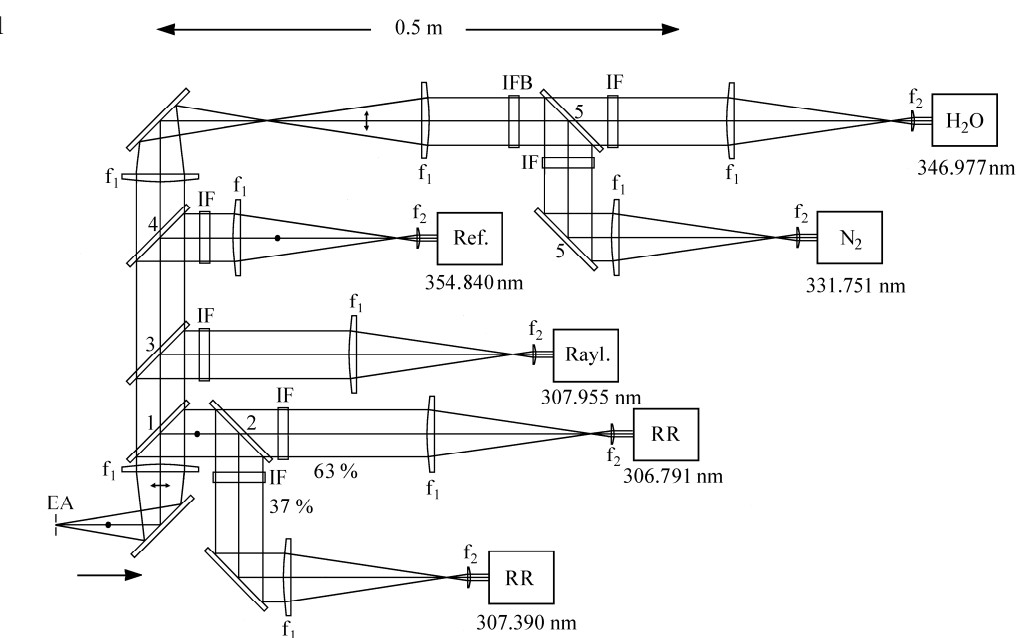

**Fig. 9.** Final polychromator design: The true orientation of the mounting plate (vertical) is rotated clockwise by
90º. The radiation cone from the telescopes (arrow) enters the polychromators from behind the plate as indicated
by the polarization dot next to the arrow. In detail:
EA:      Entrance aperture with four adjustable blades (OWIS)
1:      Beam splitter transmitting almost all P-polarized radiation (308-355 nm) and highly reflects the S-
7          polarized 308-nm radiation (Laseroptik)

2:      63 %/37 % beam splitter for S-polarized 308-nm radiation (Laseroptik)
3:      Beam splitter reflecting all radiation at 308 nm and transmitting 83-91 % of the longer-wavelength P
10          components (Laseroptik)

4:      Polarizing beam splitter (Laseroptik)
5:      Sharp-edged long-pass filter for P polarization reflecting about 99 % at 332 nm and transmitting 99 %
13          of the longer-wavelength components (Materion-Barr)

IF:      Interference filters with bandwidths of 0.25 nm except for 307.39 nm (0.15 nm) and 347 nm (0.75 nm)
15          (Materion Barr, Alluxa)

IBF:      Broadband interference filter transmitting between 330 and 355 nm with T = 85-90 % and blocking the
17          radiation outside this range by at least $10^5$ (Semrock).

Lenses: $f_1$ =150 mm and $f_2$ = 18 mm (large telescope), $f_2$ = 30 mm (small telescope)
Detailed specifications: Table 3





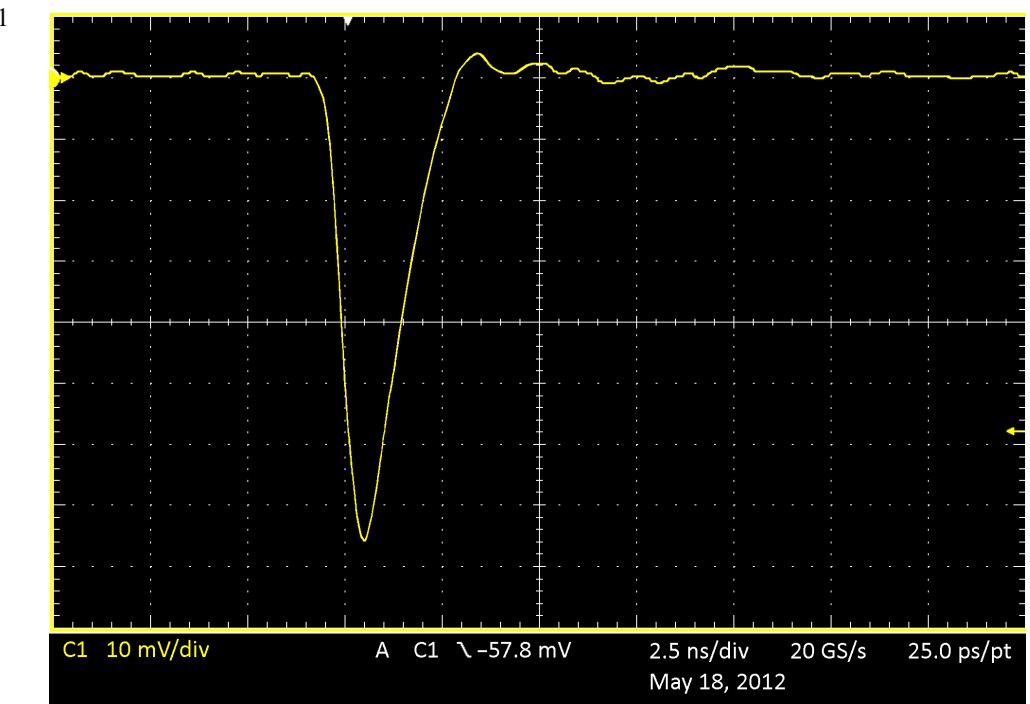

2 **Fig. 10.** Single-photon pulse from a Hamamatsu R7400P-03 PMT with the most recent version of the Romanski

3 (RSV) socket, measured with a 1-GHz digital oscilloscope (Tektronix, DPO 7104); from (Trickl et al., 2020)

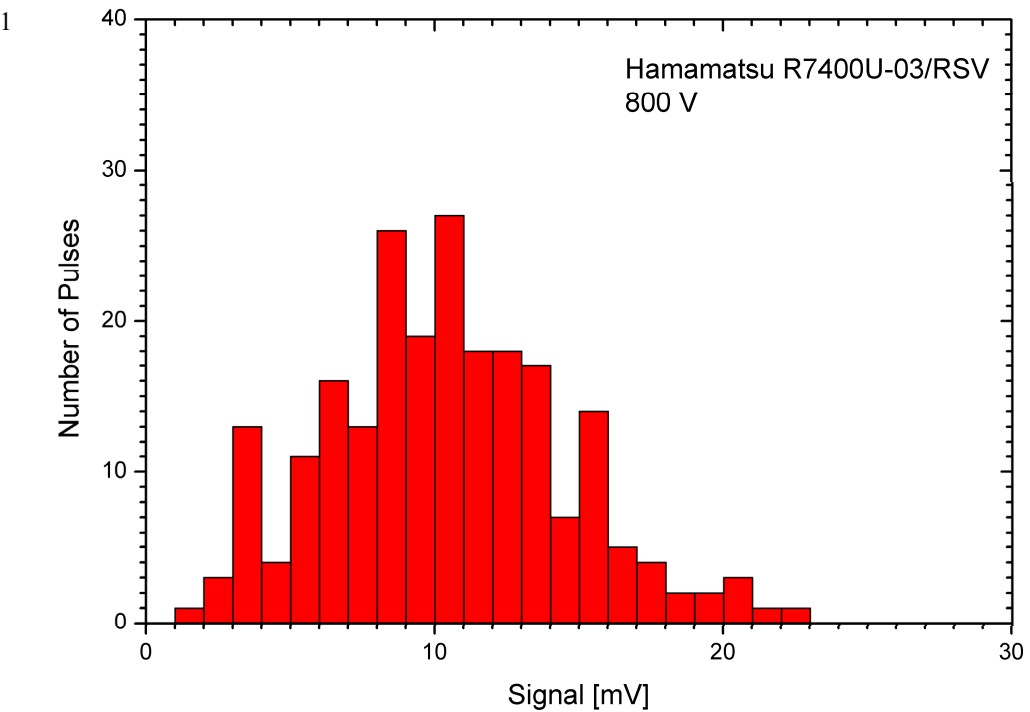

2 **Fig. 11.** Pulse height distribution of a Hamamatsu R7400-03 PMT (RSV module) for 800 V of operating voltage

3 determined from a long time scan with a 1-GHz digital oscilloscope (sign of the pulse amplitudes inverted); from

4 (Trickl, 2020).



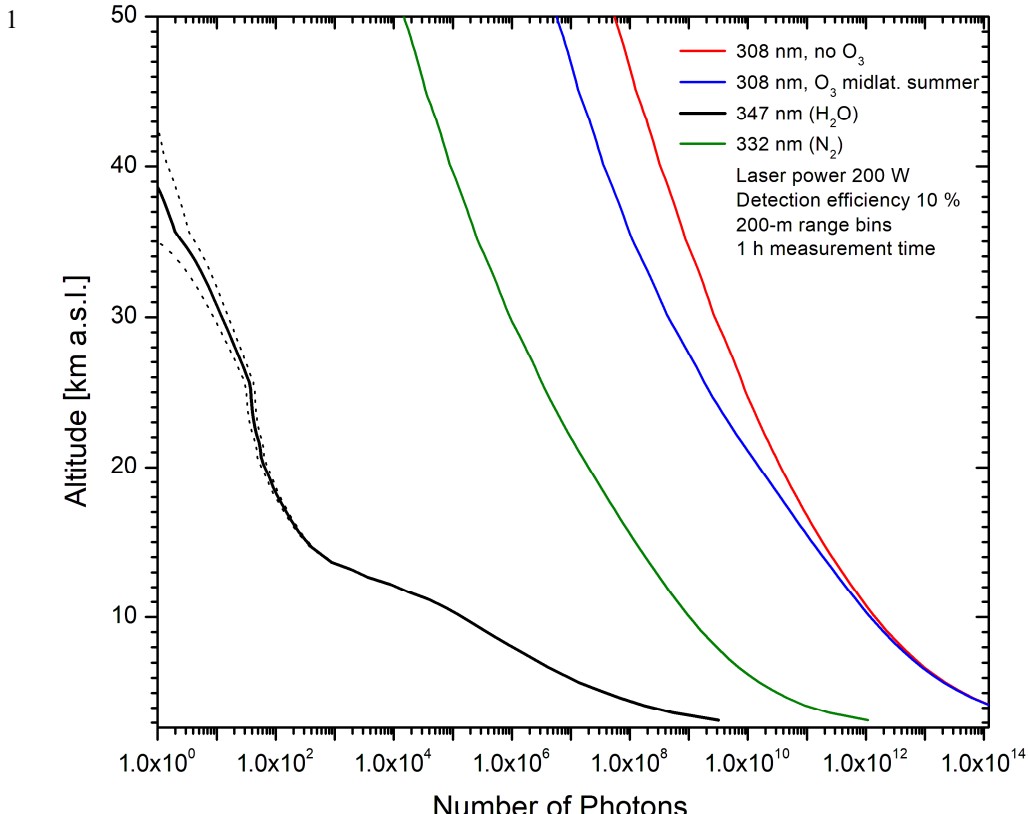

2 **Fig. 12.** Simulations of the backscatter signals for four wavelengths specified in the upper right corner; An

3 average laser power at 308 nm of 200 W, a detection efficiency of 10 %, a range bin of 200 m and a

4 measurement time of 1 h were assumed.

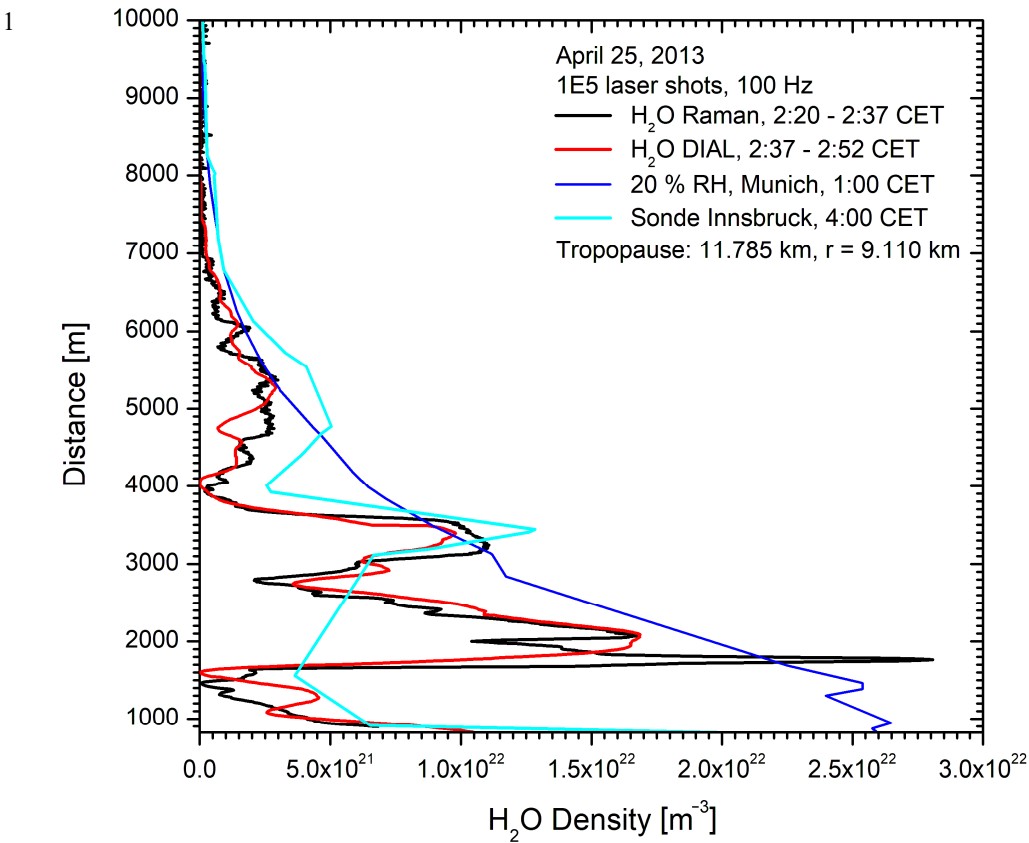

**Fig. 13.** Comparison of consecutive measurements of the Raman lidar and DIAL at UFS on 25 April, 2013: the
sonde measurements at Munich (not shown) and Innsbruck strongly differ from those of the lidar systems. For
comparison, we show the densities corresponding to 20 % RH as calculated from the Munich radiosonde.

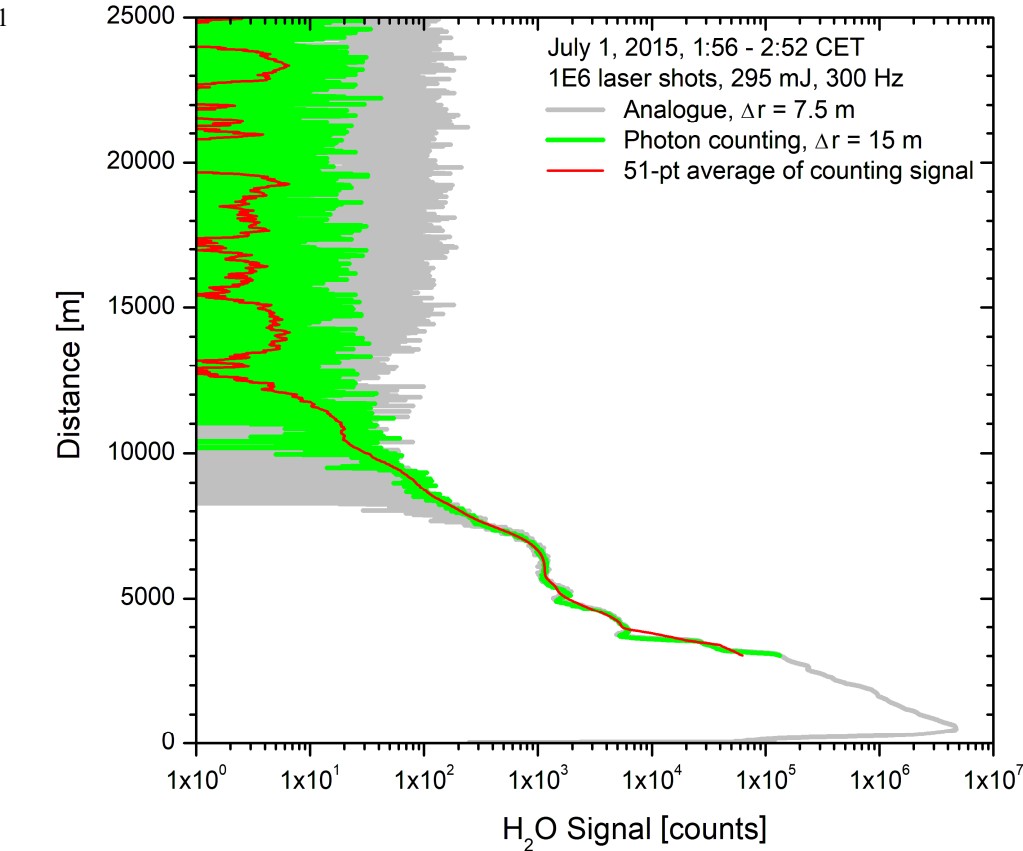

**Fig. 14.** 347-nm Raman backscatter signals as a function of the vertical distance above UFS, obtained during the
first hours on 1 July, 2015. Despite a high noise level of about 12 counts (square root of signal) the averaged
signal remains positive up to r = 19.7 km. The averaged signal covers six decades, the peak signal being roughly
3 mV. The average laser pulse energy, 295 mJ, was low due to a contaminated cell window.



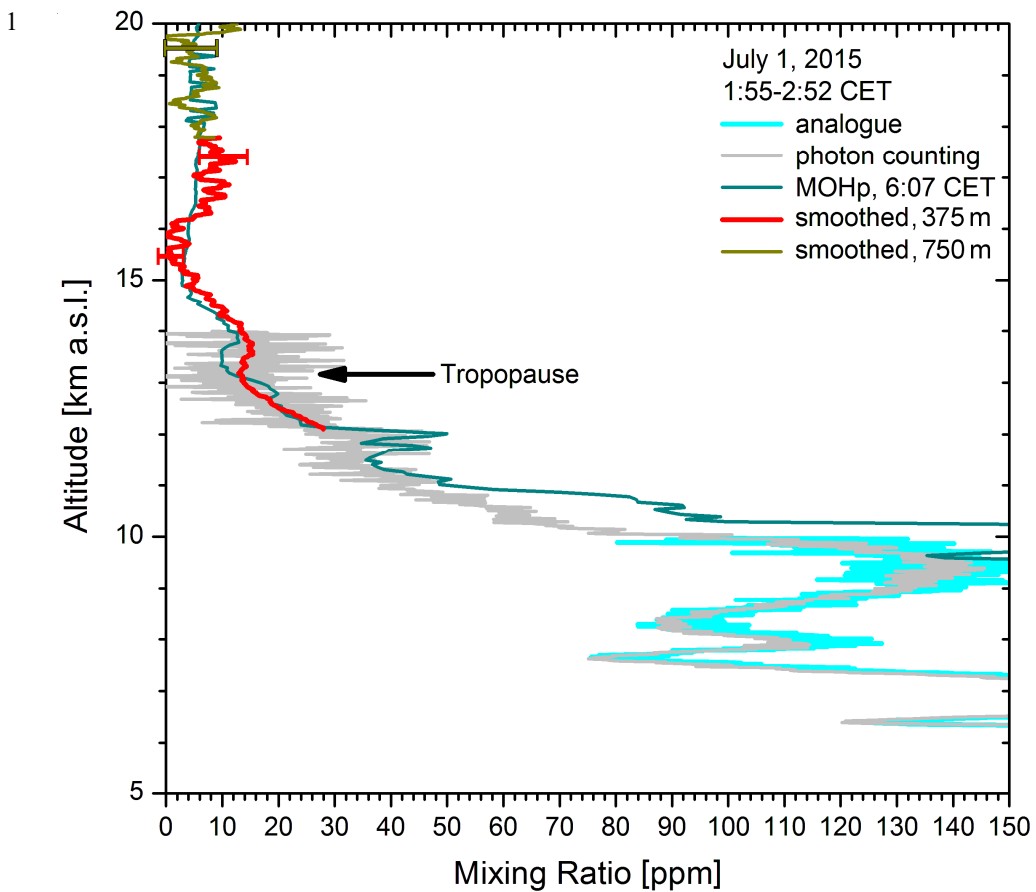

**Fig. 15.** Water-vapour mixing ratio obtained for the measurement in Fig. 15; the calibration is based on looking
at zones of best agreement below 7 km between the sonde data for Munich (1 CET), Innsbruck (4 CET) and
Hohenpeißenberg (6 CET). Just the Hohenpeißenberg (MOHp) results are displayed here because they agree best
with the lidar values above 11 km. 51-pt and 101-pt arithmetic-means smoothing was applied to the mixing
ratios derived from the photon-counting data at high altitudes, the corresponding VDI vertical resolutions are
specified in the legend.

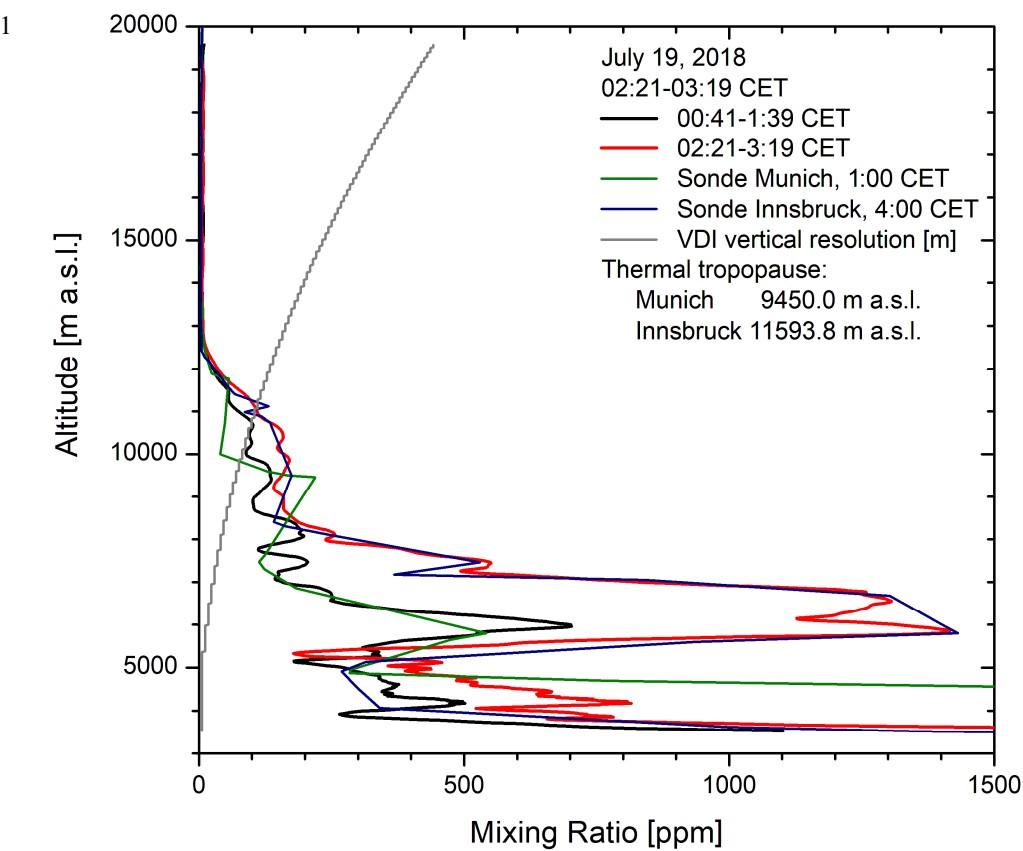

**Fig. 16.** Calibration of the measurements on 19 July 2018: The profile derived from the first measurement agrees
better with the 1:00 CET sonde data from Munich. The mixing ratios for the second measurement almost
coincides with those from the later sonde launch at the airport of Innsbruck. The average laser pulse energy was
380 mJ (300 Hz).



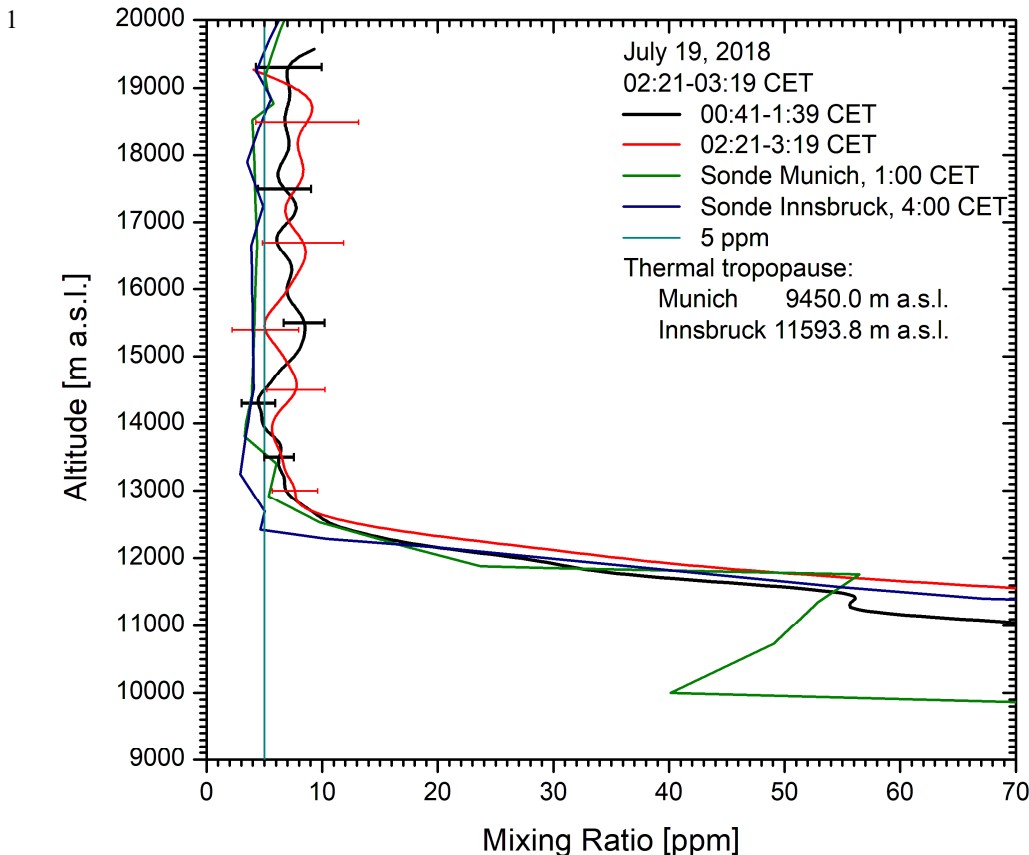

2    **Fig. 17.** Comparison of the two lidar measurements on 19 July, 2018, and the Innsbruck sonde on a zoomed

3    scale: The lidar values agree well up to 18 km, ranging between 5 ppm and 12 ppm. The mixing ratio for the

4    radiosondes (presumably RS41) is much lower than that for the lidar in the stratosphere.

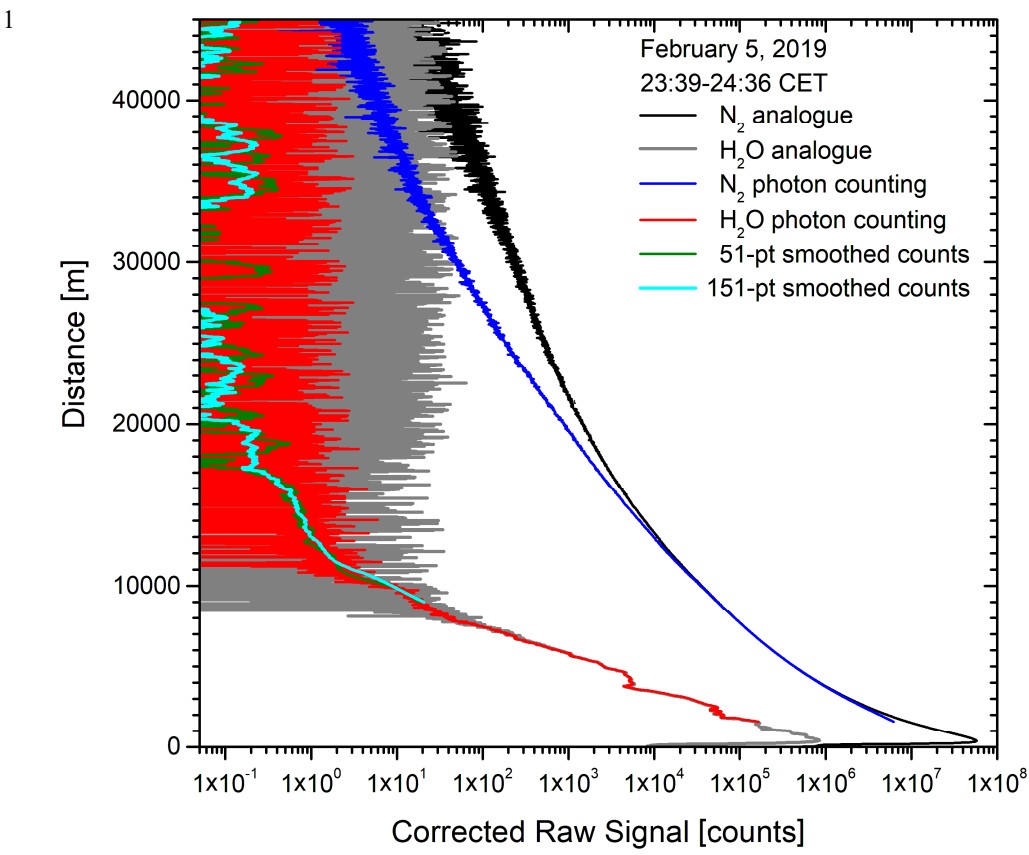

2 **Fig. 18.** Nitrogen and water-vapour backscatter signals on 5 February 2019 as a function of the vertical distance

3 above UFS; The H$_2$O backscatter profiles averaged over 151 7.5-m bins (i.e., raw data; VDI vertical resolution:

4 562.5 m) become noisy at about 17 km (19.7 km a.s.l.). The laser pulse energy was just 360 mJ (300 Hz).



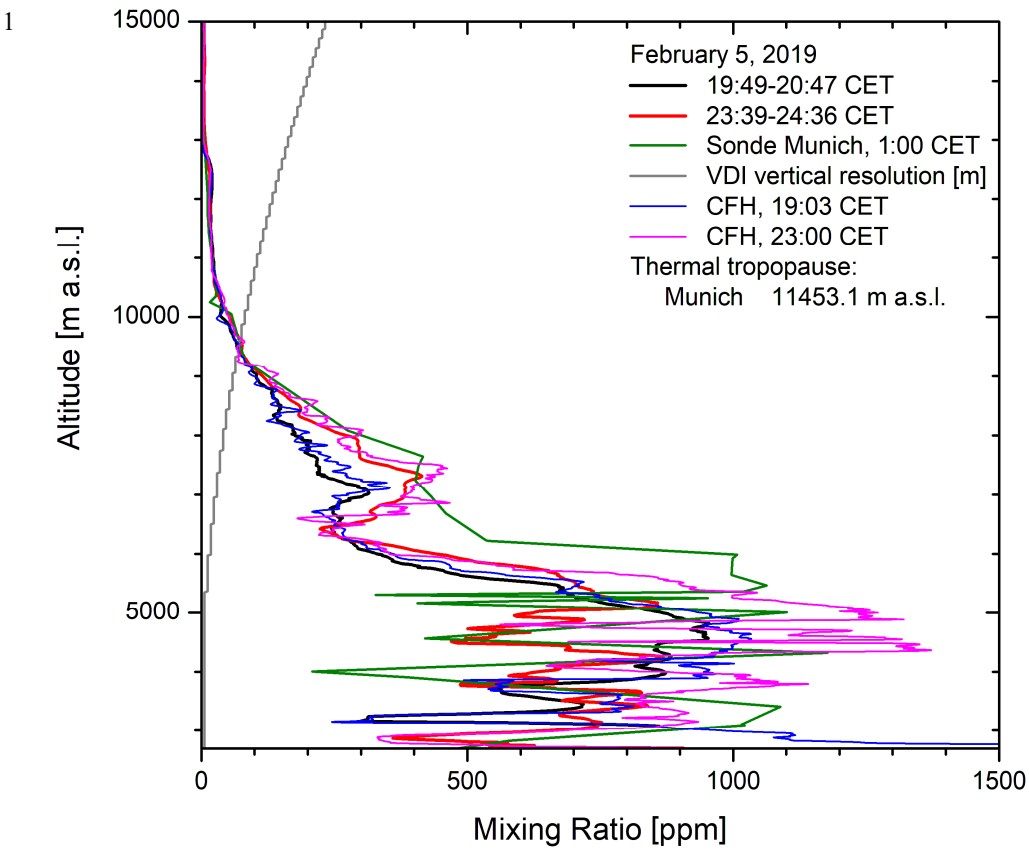

2  **Fig. 19.** Vertical distributions of water-vapour derived from two measurements of the Raman lidar on 5 February

3   2019 together with those from the midnight Munich sonde and the CFH sensors; the CFH data in the upper

4   troposphere were used for calibration.





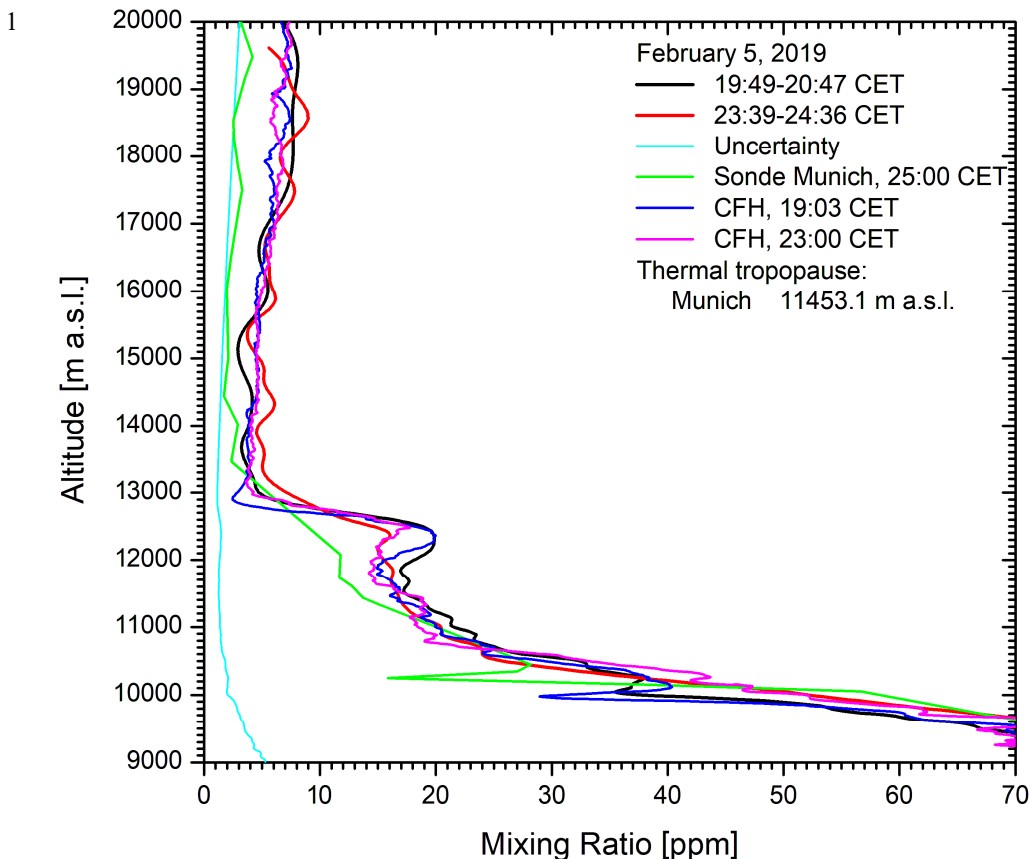

2 **Fig. 20.** Zoomed portion of Fig. 20: The agreement between lidar and CFH is satisfactory up to almost 20 km.

3 Above this, the lidar values start wider excursions around the CFH mixing ratios.

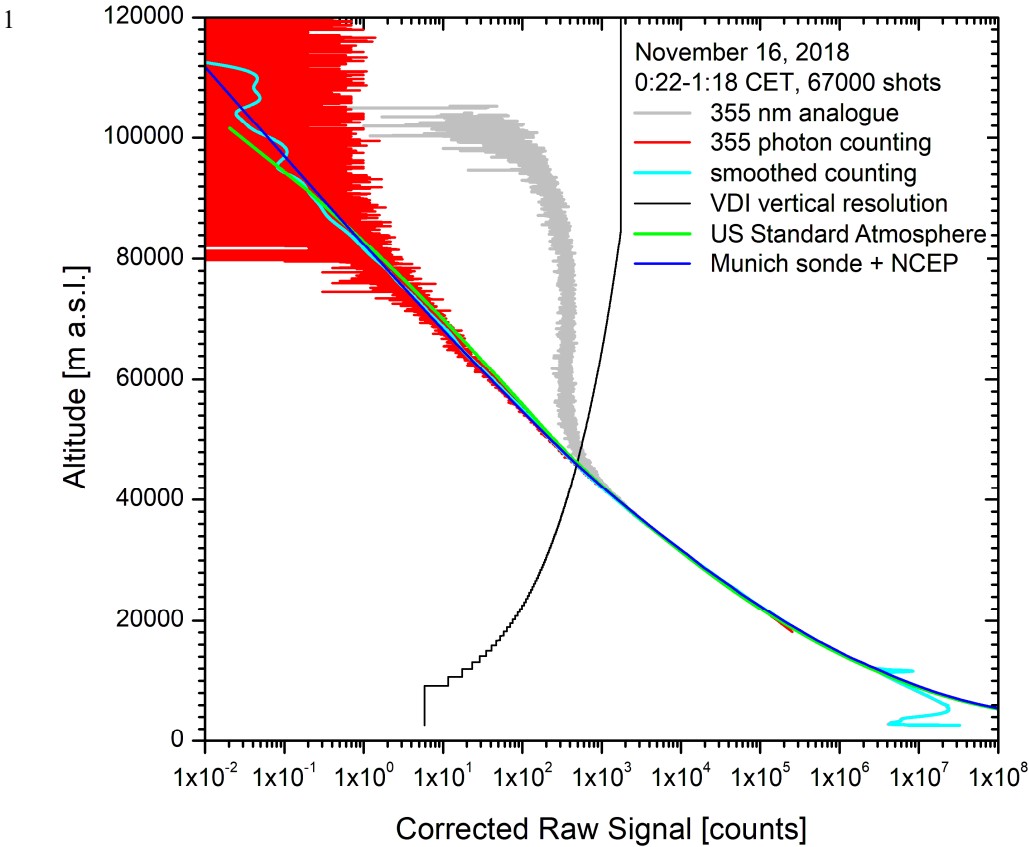

**Fig. 21.** 355-nm backscatter coefficient for a 355-nm measurement on November 16, 2018: The smoothed signal shows low noise up to almost 95 km, corresponding to more than 8 decades of signal. A simulated backscatter signal based on the U.S. Standard Atmosphere shows principal agreement, but the are some deviations. The agreement with the calculation for a combined radiosonde and NCEP profile is almost perfect up to the NCEP upper boundary of 50 km. The NCEP-based profile was extrapolated to about 100 km. The VDI vertical resolution is given in metres.

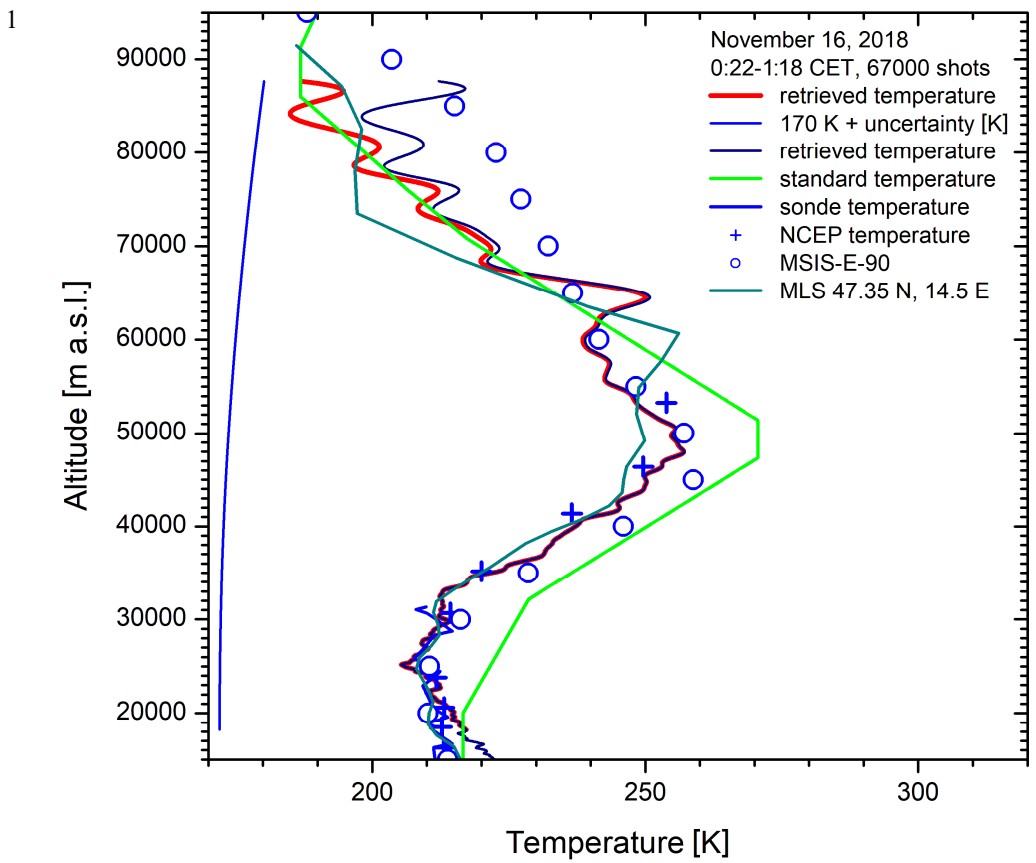

**Fig. 22.** Temperature profile from the measurement in Fig. 21, in comparison with data from the Munich 1:00 CET radiosonde, NCEP (13:00 CET), the MSIS model and MLS; the temperatures were retrieved from the lidar signal by initializing the temperature at about 87 km using both the U.S. Standard and the MSIS values. Both retrievals converge to the same curve within 15-20 km from the top.