# Peer review of "A powerful lidar system capable of one-hour measurements of 1"

_Atmospheric Measurement Techniques, 2020_

## Referee Comment (RC1) · Anonymous Referee #3 · 11 Aug 2020

Review for "A powerful lidar system capable of one-hour measurements of water vapour in the troposphere and the lower stratosphere as well as the temperature in the upper stratosphere and mesosphere"

This manuscript describes a high-power Raman lidar system has been installed at Schneefernerhaus (Garmisch-Partenkirchen, Germany) at 2675 m a.s.l., at the side of an existing wide-range differential-absorption lidar (DIAL). An industrial XeCl laser was

modified for linearly polarized single-line operation at an average power of about 180 W. This high power and a 1.5-m-diameter receiver allow us to extend the operating range for water-vapour sounding to 20 km for a measurement time of just one hour, at an uncertainty level of the mixing ratio of 1 to 2 ppm. The lidar was successfully validated with a balloon-borne cryogenic frost-point hygrometer (CFH). In addition, temperature measurements to altitudes around 87 km were demonstrated for one hour of signal averaging. The system has been calibrated with the DIAL, the CFH and radiosondes.

General Comments:

In general this manuscript describes the lidar system in an immense detail. At times it can get confusing as to which lidar system or which component is being described.

Consider alternate word choice for the term 'powerful' in the title or 'stronger' in the abstract, they are both ambiguous.

The introduction is not logically presented as many key words are missing from sentences and there is no consistent message. Consider adding a key figure or illustration that better gives the reader a perspective on where the lidar would help understand either trends or process studies (or both). There are statements without adequate referencing. This can and should be related back to the atmospheric case study to better support the importance of the vertical lidar profile.

The introduction is also quite long and some is described in the authors previous 2008 work (https://doi.org/10.1364/AO.47.002116). Consider shortening or directing the reader to this publication for more details regarding the DIAL.

In general the lidar description has very useful information but is organized as one would take field notes. There needs to be some explanation as to why many of these parameters are important. It's also not clear that the thin film polarizer or Raman cell (SRS) are actually used in the experiment as there is no final outcome described. There are too many varied parameters (focal length, rep rate, input power, cell length)

for the reader to come away with any conclusion.

Additional water vapor lidar references include : https://doi.org/10.1175/1520-0426(1995)012<1177:ACOWVM>2.0.CO;2

Technical Comments: P1L23 – disregard the first use of UTLS P1L26-29 – provide further references for these statements P1L29 – Write out NCEP in first use P3L28 – Is the lidar used in daytime or during nighttime? Or DIAL during the day and Raman system during the night?

Figure 1/3 – can these be combined?

Figure 13: How is the WV calibration calculated for this night? For instance, it looks as a scaled factor could be applied to the WV profile. What is the difference at 2km caused by?

Consider reducing the number of individual water vapor comparison profiles or make an aggregate summary plot. How many measurements of WV are there with the new system? Feb 2019 is the most compelling of the case studies as an excellent comparison with the CFH.

Conclusions: How frequent are the temperature measurements? Is this system automated?

---

## Referee Comment (RC2) · Anonymous Referee #1 · 25 Aug 2020

This manuscript describes the development of a high-power Raman water vapor lidar deployed at a high-elevation site near Garmisch-Partenkirchen, Germany. Based on its somewhat heterogeneous content and style, the manuscript seems like the concatenation of several work efforts covering the period 2012-2019. It is long and very detailed, often too detailed, which makes it sometimes difficult to follow, and also makes the actual objectives unclear. For example, the abstract mentions an "existing" co-located DIAL (used for calibration), but this system has been out of operation since 2014. An-

other example is the multiple references to temperature measurements, including in the manuscript title. These measurements are separated from the water vapour system, as they are made with a Nd:Yag laser. They do not fit well in the present manuscript, which focuses on the high-capability water vapour measurements.

Because of the level of detail and length, I strongly suggest that many sections of the manuscript be used as "Supplementary material", and that the authors leave in the main manuscript only the critical steps that led to the instrument's current configuration. The authors should add a discussion/conclusion on expectations in terms of future measurement contributions and strategic goals.

In regards to experimental development, the technical content is very important, and for that reason, should be published, assuming that Supplementary Material can be separated from the main manuscript. I therefore recommend publication after major revisions.

Suggestions for major revisions:

1) For water vapor there are only 4 measurement nights shown (4/25/2013, 7/1/2015, 7/19/2018, and 2/5/2019), including only 1 comparison with CFH. For temperature, there is only one night shown. To my opinion, this is not enough to characterize the performance of an instrument, especially when the study period spans 7 years. I strongly suggest that the authors show more profiles and/or statistics (e.g., mean differences aggregated from all available measurement nights), and possibly a climatology. Also I strongly suggest that the authors show one or more curtain plots showing the short-term (overnight) water vapor changes as a function of time and altitude. As the authors claim in the manuscript, this is one of the strengths of lidar, yet they have not shown any examples of it.

2) Below are three sections which I believe are too detailed, and therefore fit better as "Supplementary material". They can be replaced in the main manuscript by short-ened paragraphs or sentences that summarize the "take-home" message and provide

a more fluid read:

- Sections 2.2 and 2.3, and corresponding figures 2, 3, 4, 5

- Sections 3.3., 3.4, and corresponding figures 10, 11

-Section 4.4., and corresponding Fig 12 (also see my comment on this section further below)

Other major comments:

Section 4.4, and Fig 12:

This section does not bring important information to the paper, besides showing the lidar sensitivity to $H_2O$ in the UTLS. It can be removed from the manuscript. If kept in the manuscript, it should be included earlier (e.g., in the introduction) as a mean to state the problem.

Section 5 (Calibration):

Using the DIAL as a calibration source is a very interesting approach. However, it is mentioned that the DIAL has been inoperative since 2014. This section does not provide any other information on how the Raman lidar has been calibrated since then (including accuracy), and most importantly, what is the timeline to use the DIAL system again for calibration (if it will ever be used again).

Minor comments:

P3/l2, Radiosonde weakness: Outdated literature. RS92 has been replaced by RS41, with better performance in the UTLS

P3/l4, CFH: Use full instrument name Cryogenic Frost-Point Hygrometer

P3/l8, "Raman scattering is a background-free method": this is an awkward expression. Please rephrase.

P3/l33, DIAL accuracy: Proper reference should point to HITRAN 2008. Cited references mention 2% and 3.2%. Please clarify where the figure "1% or less" comes from.

P3/l37, "355 nm" and p4/l2: The single temperature profile shown in this manuscript comes from a Nd:Yag, not the Excimer laser described for water vapour. This distinction makes the inclusion of temperature in this manuscript somewhat off-subject.

P7/l31 and l33: Should refer to fig 7, not fig 6.

P13/l8, "multiplying the backscatter signal …. by the square of distance r": If both the N2 and H2O detectors are triggered simultaneously, there is no need for range correction at all, which makes this sentence out of context. Please clarify or modify the sentence.

P13/l15, "the influence of ozone exactly cancels because the transmitted wavelength is the same": This is an incorrect statement. There is a slight extinction differential associated with the returned signals (after backscatter). However, it is expected to be small for most of the troposphere because absorption at 332 and 347 nm is weak. It can however impact the measurement in the LS as ozone becomes more abundant. Please provide estimate of the ozone interference (this is quantifiable)

P13/l18, "data are collected at 51.2 ns per bin": Is the laser pulse length 80 ns? What are the implications of such a long pulse on data sampling?

P13/l39, "The role of aerosols is limited to extinction in a Raman lidar": Not entirely true. Fluorescence has been observed as well (e.g., Immler et al., 2005). Please modify sentence accordingly and discuss its impact. For example, a strong, widespread and sustained episode of bio-fluorescence was observed at Northern mid-latitudes during Fall 2017 after the so-called Chisolm PyroCb event. Did the IMK-IFU lidar observe it?

P14/l31, "geopotential altitudes into real ones": Replace "real" by "geometric"

P15/l15, uncertainty equation, and p15/l19: This is a quite arbitrary estimate. There are multiple references detailing direct computation of temperature uncertainty using

the density integration technique (including Hauchecorne and Chanin, 1980 for random uncertainty; the Klett references cited, and Leblanc et al., 2016). Why not using those? Please clarify/expand.

P17 and fig 14: These results have been shown before at several occasions. The lengthy discussion is unnecessary

P18/l19,"not as reliable": What do the authors mean by "not as reliable"? What is the physical basis for such statement?

P18/l20, "We speculate that this is due to a change in sonde type from RS 92 to RS 41": RS41 is expected to be more accurate than RS92 in the LS. Please justify this statement that seems to go against current general thinking about the latest RS41 radiosondes

P19, section 6.2: The entire section, together with Fig. 21, is of poor scientific or technical interest. I would like to suggest to just remove it altogether. One of the correlative curves is extrapolated, which has no scientific value, and the apparent agreement between the lidar and correlative densities is fully expected with a figure of such aspect ratio (showing 10 orders of magnitude on the X-axis!). If any relative density comparison really must be shown, please use a ratio to the US standard atmosphere density for better visibility.

P21/l25, "The temperature measurements ... were quite successful": Only one temperature profile is shown to support this conclusion. Please modify to mitigate this statement, or include more temperature comparisons to support it.

Syntaxing and formatting issues:

p1/l23: UTLS

p1/l33: Missing verb

p1/l36: Syntax

p2/l36:: Add "water vapor" between "ground-based" and "lidar"

p17/l38, "background signal": missing verb

p18/l24: Syntax

p27: Trickl/Wilson references misplaced

p28: Two conflicting 2020 references for Trickl 2020

Figures:

A lot of figure captions contain excessive narrative discussion. Please move those to the main text. Fig 9 caption: way too long and detailed. Most of it should be kept in the main text or put together in a table

Figure 21 caption: too long. Keep discussion in the main text.
* * *

---

## Author Comment (AC1) · 26 Oct 2020

**Reply to the reports on manuscript AMT-2020-90**

Thomas Trickl, October 26, 2020

The comments received are highly valuable. Indeed, shortening is reasonable. Since the paper was written over quite a few years the importance of some of the technical has changed. The focus must be on the most recent state of knowledge.

The development and testing of the system has undergone several stages, interrupted by waiting for new equipment when work for other projects was done by our small team. This is the reason why we just recently reached a technical and data-evaluation level to start routine measurements.

The text of the two reports are given in italics, the replies normal. A file with the changes marked is submitted as supplementary material.

**Review 1:**

We appreciate the particularly careful reading of our paper. This has been very helpful indeed!

*This manuscript describes the development of a high-power Raman water vapor lidar deployed at a high-elevation site near Garmisch-Partenkirchen, Germany. Based on its somewhat heterogeneous content and style, the manuscript seems like the concatenation of several work efforts covering the period 2012-2019. It is long and very detailed, often too detailed, which makes it sometimes difficult to follow, and also makes the actual objectives unclear. For example, the abstract mentions an "existing" co-located DIAL (used for calibration), but this system has been out of operation since 2014. Another example is the multiple references to temperature measurements, including in the manuscript title. These measurements are separated from the water vapour system, as they are made with a Nd:Yag laser. They do not fit well in the present manuscript, which focuses on the high-capability water vapour measurements.*

Our goal has been to describe the lidar system as a whole. The entire receiver design is made for water vapour, ozone and temperature. This must be explained. In particular, one chapter is devoted to characterizing the capability for temperature measurements. The Nd:YAG laser is part of the system, similar to current implementations of stratospheric ozone DIALs at several stations!!!

Of course, water vapour is our primary target and most of the paper is about $H_2O$. However, when the system was funded there was a strong demand also on T measurements by local groups.

The long T part in the title emerged from a discussion with the editor. Initially, the title was much shorter.

The DIAL was used in a demonstration of a side-by-side calibration of the Raman lidar. A new laser system for this DIAL is under development, but this development has proceeded slowly recently due the current lack of personnel, especially after my retirement. Emission was already observed for a much faster repetition rate of 100 Hz. The future use of the DIAL is promised in the conclusions.

*Because of the level of detail and length, I strongly suggest that many sections of the manuscript be used as "Supplementary material", and that the authors leave in the main manuscript only the critical steps that led to the instrument's current configuration. The authors should add a discussion/conclusion on expectations in terms of future measurement contributions and strategic goals.*

During the revision I tried to shorten the content as far as possible. I personally dislike "supplementary material" because this means additional downloading efforts for the reader. The length of paper is really not excessive.

A few statements about the future were added.

*In regards to experimental development, the technical content is very important, and for that reason, should be published, assuming that Supplementary Material can be separated from the main manuscript. I therefore recommend publication after major revisions.*

*Suggestions for major revisions:*

1) *For water vapor there are only 4 measurement nights shown (4/25/2013, 7/1/2015, 7/19/2018, and 2/5/2019), including only 1 comparison with CFH. For temperature, there is only one night shown. To my opinion, this is not enough to characterize the performance of an instrument, especially when the study period spans 7 years. I strongly suggest that the authors show more profiles and/or statistics (e.g., mean differences aggregated from all available measurement nights), and possibly a climatology. Also I strongly suggest that the authors show one or more curtain plots showing the shortterm (overnight) water vapor changes as a function of time and altitude. As the authors claim in the manuscript, this is one of the strengths of lidar, yet they have not shown any examples of it.*

The measurements were selected from a much larger number starting in 2018 as specified in the first paragraph of Sec. 6.1. All these profiles look reasonable. The examples were chosen to show the performance under different conditions, such as high and low noise or different situations and methods for the calibration. An introductory remark was added.

This is not a paper on system validation: It is paper on the system development and testing. A full-size validation exercise has been out of scope and would require additional funding. Unfortunately, the CFH team was present just for two days, the second day being devoted to cloud cases!

*2) Below are three sections which I believe are too detailed, and therefore fit better as "Supplementary material". They can be replaced in the main manuscript by shortened paragraphs or sentences that summarize the "take-home" message and provide a more fluid read:*

*- Sections 2.2 and 2.3, and corresponding figures 2, 3, 4, 5*

These sections are now shortened and two figures were removed. There are paragraphs that were important at some time in the project, but no longer matter. We spent significant time on these studies and on buying and testing new components which explains the amount of detail! Now, just the outcome of these efforts is briefly described (e.g., spectral behaviour, Raman scattering). The chapter on stimulated Raman scattering now just explains why this approach was abandoned, an important message for future efforts elsewhere.

*- Sections 3.3., 3.4, and corresponding figures 10, 11*

We believe that these section are key sections and should not serve as supplementary material. This is a paper on the system development and testing, and this kind of information is the basis for the understanding of the results discussed. However, considering the rather complete description in our companion paper on our tropospheric ozone DIAL systems, some details were dropped.

*-Section 4.4., and corresponding Fig 12 (also see my comment on this section further below)*

*Other major comments:*

*Section 4.4, and Fig 12:*

*This section does not bring important information to the paper, besides showing the lidar sensitivity to H2O in the UTLS. It can be removed from the manuscript. If kept in the manuscript, it should be included earlier (e.g., in the introduction) as a mean to state the problem.*

This chapter is needed to judge the results (see discussion section). In the introduction it would be too early. It is short anyway.

*Section 5 (Calibration):*

*Using the DIAL as a calibration source is a very interesting approach. However, it is mentioned that the DIAL has been inoperative since 2014. This section does not provide any other information on how the Raman lidar has been calibrated since then (including accuracy), and most importantly, what is the timeline to use the DIAL system again for calibration (if it will ever be used again).*

The laser development for the DIAL looks very promising, and it is planned to complete it. This is clearly stated in Sec. 5 and in the conclusions. I added a paragraph on the sondes to Sec. 5.

*Minor comments:*

*P3/l2, Radiosonde weakness: Outdated literature. RS92 has been replaced by RS41, with better performance in the UTLS*

Our results show that RS 41b does not mean any improvement at all in the lower stratosphere. Maybe the sonde itself is better than RS 92. However, the corrections for RS 92 seem to be better. As mentioned in Sec. 6 there is also a wet bias for low RH values in the troposphere that does not exist in the RS 92 profiles. I removed the sentence on the RS 92 sonde: Too much detail in the introduction!

*P3/l4, CFH: Use full instrument name Cryogenic Frost-Point Hygrometer*

Done, next occurrence of CFH on P. 3 also modified,

*P3/l8, "Raman scattering is a background-free method": this is an awkward expression. Please rephrase.*

This is an expression frequently used in laser-based analytics. However, I slightly modified it.

*P3/l33, DIAL accuracy: Proper reference should point to HITRAN 2008. Cited references mention 2% and 3.2%. Please clarify where the figure "1% or less" comes from.*

This comes from our own validation exercises in the papers cited! Also a comparison with HITRAN is made there. The spectroscopic data of Ponsardin and Browell, used for the calibration of our DIAL, are obviously more accurate.

*P3/l37, "355 nm" and p4/l2: The single temperature profile shown in this manuscript comes from a Nd:Yag, not the Excimer laser described for water vapour. This distinction makes the inclusion of temperature in this manuscript somewhat off-subject.*

Numerous temperature profiles were retrieved from measurements with the Raman-shifted excimer which will be in part presented by L. Klanner in her thesis.

*P7/l31 and l33: Should refer to fig 7, not fig 6.*

Thank you! The numbers have changed anyway due to discarding two figures.

*P13/l8, "multiplying the backscatter signal : : :. by the square of distance r": If both the N2 and H2O detectors are triggered simultaneously, there is no need for range correction at all, which makes this sentence out of context. Please clarify or modify the sentence.*

This sentence is correct: The quantity discussed in this statement is the number density, for me the primary target quantity. The statement makes clear that (if ozone is neglected) the data evaluation is simple and robust. However, for the test period presented here we preferred to avoid an ozone correction due to the problems with the generation of 353-nm radiation. I added a remark further below.

*P13/l15, "the influence of ozone exactly cancels because the transmitted wavelength is the same": This is an incorrect statement. There is a slight extinction differential associated with the returned signals (after backscatter). However, it is expected to be small for most of the troposphere because absorption at 332 and 347 nm is weak. It can however impact the measurement in the LS as ozone becomes more abundant. Please provide estimate of the ozone interference (this is quantifiable)*

Thank you for this remark! Indeed, a small deviation of exists in the $N_2$ signals that reached almost 2 % at 20 km, estimated from the ozone climatology of Hohenpeißenberg for the months with maximum ozone. A sentence pointing this out was added. The ozone influence in the $H_2O$ channel is lower by one order of magnitude.

*P13/l18, "data are collected at 51.2 ns per bin": Is the laser pulse length 80 ns? What are the implications of such a long pulse on data sampling?*

Not more than 2 bins! In the UTLS smoothing is applied. I think we should not comment on this issue. For near-field data the exact position of the zero point matters indeed.

*P13/l39, "The role of aerosols is limited to extinction in a Raman lidar": Not entirely true. Fluorescence has been observed as well (e.g., Immler et al., 2005). Please modify sentence accordingly and discuss its impact. For example, a strong, widespread and sustained episode of bio-fluorescence was observed at Northern mid-latitudes during Fall 2017 after the so-called Chisolm PyroCb event. Did the IMK-IFU lidar observe it? Please clarify/expand.*

The work of Franz Immler and Jens Reichardt is well known to me, thank you for reminding me. Our measurements were all made during clear nights without aerosol over many hours to allow for playing with the settings. I added two statements.

*P17 and fig 14: These results have been shown before at several occasions. The lengthy discussion is unnecessary*

These results were to some extent explained in the 2017 ILRC proceedings. I think it is not fair just to refer to that paper. The figures are presented now in a much better way, including smoothing, and show more detail. The calibration figure in the ILRC paper is not repeated (now cited).

I took this example because since it shows what can be achieved under worst-case conditions, i.e., with an open polychromator entrance slit with strong lunar background, that stratospheric water vapour can be resolved even under these conditions. In addition, there is good agreement with an RS 92 sonde (in contrast to that with RS 41 in the other examples). From a scientific point of view, the case is interesting because of a very dry layer, which indicates that the distributions do not look the same all the time in the stratosphere.

The section is not really long. Some modifications were made.

*P18/l19,"not as reliable": What do the authors mean by "not as reliable"? What is the physical basis for such statement?*

I modified that part.

*P18/l20, "We speculate that this is due to a change in sonde type from RS 92 to RS 41": RS41 is expected to be more accurate than RS92 in the LS. Please justify this statement that seems to go against current general thinking about the latest RS41 radiosondes*

We give several comparisons in this section suggesting a non-ideal behaviour of RS 41. The statement about our tropospheric studies underlines that there are some issues with RS 41 that do not exist for RS 92. I informed the German Weather Service about this. Maybe the RS 41 sonde itself is better than RS 92, but adequate corrections are missing.

*P19, section 6.2: The entire section, together with Fig. 21, is of poor scientific or technical interest. I would like to suggest to just remove it altogether. One of the correlative curves is extrapolated, which has no scientific value, and the apparent agreement between the lidar and correlative densities is fully expected with a figure of such aspect ratio (showing 10 orders of magnitude on the X-axis!). If any relative density comparison really must be shown, please use a ratio to the US standard atmosphere density for better visibility.*

From the remarks I conclude that not the entire section 6.2 is meant, just Fig. 21 and its description. This is an exclusively technical paper. The scientific importance is described in the introduction. In Fig. 21 (19), we give the best example for the system performance, the scale extending out to 120 km. with average counts below 1. It is generally the habit in lidar papers to present examples of the raw data.

Indeed, the NCEP data end at 50 geopotential km. This was respected in Fig. 22. Here, the extrapolation was made to guide the eyes. I remove the extrapolation.

It is a good suggestion to show that ratio, but the details can be also (and better) seen in Fig. 22.

Some adjustments were made.

*P21/l25, "The temperature measurements : : : were quite successful": Only one temperature profile is shown to support this conclusion. Please modify to mitigate this statement, or include more temperature comparisons to support it.*

Done.

*Syntaxing and formatting issues:*

*p1/l23: UTLS*

Removed.

*p1/l33: Missing verb*

"shows" added.

*p1/l36: Syntax*

Changed.

*p2/l36:: Add "water vapor" between "ground-based" and "lidar"*

Added.

*p17/l38, "background signal": missing verb*

"and" removed.

*p18/l24: Syntax*

Second "5" replaced by "6".

*p27: Trickl/Wilson references misplaced*

The paper co-authored by K. Wilson is not misplaced. Also "(1998)" refers to a different paper. However, I introduced 2010a and 2010b.

*p28: Two conflicting 2020 references for Trickl 2020*

Also here I introduced a and b.

*Figures:*

*A lot of figure captions contain excessive narrative discussion. Please move those to the main text. Fig 9 caption: way too long and detailed. Most of it should be kept in the main text or put together in a table*

I found several sentences or phrases that could be removed, in particular narrative ones. The captions now explain just what is seen in the figures (e.g. the abbreviations), which is the habit.

*Figure 21 caption: too long. Keep discussion in the main text.*

Indeed, the text is narrative: Changed!

**Review 3:**

*This manuscript describes a high-power Raman lidar system has been installed at Schneefernerhaus (Garmisch-Partenkirchen, Germany) at 2675 m a.s.l., at the side of an existing wide-range differential-absorption lidar (DIAL). An industrial XeCl laser was modified for linearly polarized single-line operation at an average power of about 180W. This high power and a 1.5-m-diameter receiver allow us to extend the operating range for water-vapour sounding to 20 km for a measurement time of just one hour, at an uncertainty level of the mixing ratio of 1 to 2 ppm. The lidar was successfully validated with a balloon-borne cryogenic frost-point hygrometer (CFH). In addition, temperature measurements to altitudes around 87 km were demonstrated for one hour of signal averaging. The system has been calibrated with the DIAL, the CFH and radiosondes.*

*General Comments:*

*In general this manuscript describes the lidar system in an immense detail. At times it can get confusing as to which lidar system or which component is being described.*

The laser section was shortened as indicated in the first reply (see above).

*Consider alternate word choice for the term 'powerful' in the title or 'stronger' in the abstract, they are both ambiguous.*

The title is already rather long. It does not make sense to add complexity. "Powerful" is underlined by the measurement time of 1 h. The rest is explained in the paper.

"stronger" is explained by the vertical resolution specified in the first part of the sentence. However, I deleted the second part because the message is trivial.

*The introduction is not logically presented as many key words are missing from sentences and there is no consistent message. Consider adding a key figure or illustration that better gives the reader a perspective on where the lidar would help understand either trends or process studies (or both). There are statements without adequate referencing. This can and should be related back to the atmospheric case study to better support the importance of the vertical lidar profile.*

Which key words are missing?

*The introduction is also quite long and some is described in the authors previous 2008 work (https://doi.org/10.1364/AO.47.002116). Consider shortening or directing the reader to this publication for more details regarding the DIAL.*

The length of the introduction is rather typical in comparison with other full-size papers. It covers a reasonable number of aspects relevant for wide-range sounding of water vapour.

I do not understand the remark about the DIAL at all: That paragraph is already short and the reader is lead to the relevant papers as demanded.

*In general the lidar description has very useful information but is organized as one would take field notes. There needs to be some explanation as to why many of these parameters are important. It's also not clear that the thin film polarizer or Raman cell (SRS) are actually used in the experiment as there is no final outcome described. There are too many varied parameters (focal length, rep rate, input power, cell length) for the reader to come away with any conclusion.*

We removed complexity. The Raman-shifting chapter was shortened since Raman shifting was abandoned. It was not completely removed since whoever wants to build such a system should know that this approach has its limits.

*Additional water vapor lidar references include : https://doi.org/10.1175/1520-0426(1995)012<1177:ACOWVM>2.0.CO;2*

This and other papers are known to me. Here, we just cite lidar systems for with demonstrated a range exceeding the tropopause was demonstrated (LS in UTLS).

*Technical Comments:*

*P1L23 – disregard the first use of UTLS*

Done.

*P1L26-29 – provide further references for these statements*

There is a host of literature on transport into the UTLS region. Some of it was discussed in our STT paper 2014. Most of the investigations a related to airborne measurements, especially by the MOZAIC and CARIBIC teams. Because of the typical flight levels the information is mostly limited to the tropopause region, where a "mixing layer" exists. In our aerosol measurements (Trickl et al., 2013) we frequently see aerosol contributions around the tropopause, related to moderate volcanic eruptions, fire plumes or desert dust.

Transport farther into the stratosphere is reported for aerosol observations (e.g., Fromm et al. in various publications). Immler et al. (2005) show one case of fluorescing aerosol that was very likely lifted to altitudes beyond the tropopause region in a WCB outflow. However, the sensitivity of that

lidar for water vapour in the stratosphere was limited, also because of the co-incident fluorescence peak.

Transport to the tropopause region remains interesting. Transport to altitudes beyond the UTLS seems to be an open territory. In any case: this paper is fully technical and the description of scientific issues should remain limited. I added a few more references on key papers on the chemical composition of the UTLS layer.

*P1L29 – Write out NCEP in first use*

Done.

*P3L28 – Is the lidar used in daytime or during nighttime? Or DIAL during the day and Raman system during the night?*

I added "during night-time" to the sentence about the Raman lidar. It is obvious that the DIAL must also be operated during night-time to achieve a calibration, in addition to day-time sounding.

*Figure 1/3 – can these be combined?*

I do not see a reason for this. This would blow up the caption.

*Figure 13: How is the WV calibration calculated for this night? For instance, it looks as a scaled factor could be applied to the WV profile. What is the difference at 2km caused by?*

Calibration of course means determining an average scale factor. As pointed out this was done at higher altitudes where the influence of the artefact mentioned was less severe. The result looks acceptable also at lower altitudes.

2 km: As mentioned the measurements for the two lidar systems were not made exactly at the same time. Changes in humidity may occur at very short time scales (Vogelmann, 2011). I added a statement concerning this.

*Consider reducing the number of individual water vapor comparison profiles or make an aggregate summary plot. How many measurements of WV are there with the new system? Feb 2019 is the most compelling of the case studies as an excellent comparison with the CFH.*

Here, we just give a description on the system development and some idea about the performance. The figures have been chosen from the aspects of the limitations of the comparison with sondes and the necessity for DIAL-based calibration. I think the selection of figures is adequate.

*Conclusions: How frequent are the temperature measurements? Is this system automated?*

This system will soon be automated or, at least, will be operated under remote control via the internet, as indicated in the text. Most options have been tested. A problem is the photon-counting systems that, for reliable starts of the measurements, must be operated via mouse click (on a remote computer). The NDACC aerosol measurements have been performed at UFS in this way in recent years.

A statement was added to the conclusions.

---

## Author Comment (AC2) · 26 Oct 2020

Common file for both reviews already sent in first reply.
* * *

---

## Author Response (AR2)

**Reply to the final report on manuscript AMT-2020-90**

Thomas Trickl, November 28, 2020

**Reply to Referee 1:**

*Although some efforts have been made to address some of the reviewers' concerns about length and detail, to my opinion the revised version remained somewhat at a status quo. Two reviewers have noted that the manuscript was too detailed, to a point that the reader eventually can lose focus, not foreseeing the actual objectives of these instrument development efforts. My suggestion to move some content into Supplementary material was simply rejected by the authors. Also, several figures that I recommended to just move out or delete are still in version 4. Two examples are the unnecessary Fig 8 and 9, which are already published elsewhere.*

In principle, I dislike moving material into supplementary material since this means an additional effort for the reader. An appendix is better, but I do not see what should be moved there without interrupting the flow. Nevertheless, I agree to shortening and citing already published work where the content is not crucial for the final system design, which was done in several sections.

Indeed, Figs. 8 and 9 are part of the paper on our ozone DIAL systems that is now in press and, thus, available. I removed them, expanding the description.

In principle, one could remove the section on Raman shifting and just describe the final solution. I decided to keep it since Raman shifting has been the standard solution in many ozone DIAL systems and since I want to clarify why we finally decided to abandon it. Again, I shortened the section and referred to two ILRC contributions. Diagrams will be published by Lisa Klanner in her thesis which in under revision and, thus, cannot be cited here.

*All in all, my opinion is that not enough efforts were made to shorten and refocus the content of the manuscript to the description of a "live" instrument with clear scientific objectives. These things said, I understand that in some cases, detailed descriptions of experimental development can be a good thing, and this might be an acceptable path forward.*

In the new version I tried to eliminate (as far as possible) material not needed for understanding the final system design. However, this is a technical paper, and I think that technical details are important given the complexity of the development. It is obvious that there will be more of these powerful systems in the future. It is not only the laser system that is novel. Also the performance of the detection electronics under these conditions has meant a step into unknown territory for us.

*I therefore recommend further minor revisions and recommend the manuscript to be accepted after the following revisions are made:*

1) *Shorten section 3.4, remove Figs 8 and 9. Refer to original figures and text published elsewhere, and shorten the corresponding section so that only the type of equipment and final performance is mentioned. There is no need to repeat such technical details already published elsewhere.*

Done.

2) *Move section 4.4 (lidar signal simulation) BEFORE the instrumental development sections. This section/paragraph and the corresponding Fig 10 can be included at the end of the introduction, or as its own section, to introduce the technical requirements of the system. The authors themselves state "Before finalizing the lidar design a number of simulations of the system performance were made", clearly showing that it is more logical to discuss the simulations first*

*as a mean to determine the required specifications of the system, these latter being detailed afterwards in the manuscript.*

Done: this is a reasonable solution!

***Reply to Editor´s Comments:***

*Comments to the Author:*

*I would like you to make the following changes in the manuscript, which I believe will make the manuscript more valuable to the community as a reference for your system.*

*Both referees requested certain detailed material either be deleted or moved to a supplement or Appendix. Would you please look over the comments again and consider moving or removing some of it? I would suggest moving it which isn't hard, you could just move some of the paragraphs to the supplementary material section.*

*I have separately sent some specific changes to address.*

*Non-public comments to the Author:*

*Specific changes requested*

*1. Please remove Figures 8 and 9, and just cite where they have already appeared. Please shorten section 3.4 so that only the type of equipment and final performance is mentioned. There is no need to repeat technical details published previously, just reference it.*

*2. Please move section 4.4 (lidar signal simulation) so it is before the instrumental development sections. This section/paragraph and the corresponding Fig 10 can be included at the end of the introduction, or as its own section, to introduce the technical requirements of the system early on. This change would help the reader understand design choices described later.*

*3. Please go back over the original referee's reports and reconsider if you can make more of their changes, in particular moving sections to the Supplement section. I won't require this, but I personally agree with many of their points and think that changing them would improve the manuscript by making it easier to follow.*

I carefully re-examined the manuscript and made some improvements as described above. Instead of moving material into a supplementary section I cited our parallel paper on the ozone lidar systems or earlier ILRC work. Several paragraphs, details and sentences were removed on order to focus as far as possible just on the most recent design. I also found several ways to make the text more fluent.

To my opinion a certain amount of detail must be contained in a full-size technical paper like this one. This system development has brought a lot of new experience that should be documented. The detectors (that were introduced to the lidar community by us, together with the Licel electronics in 1995) have never been fully characterized before the current two lidar papers. Some of the findings must be outlined to understand the results for this big system.

As in the previous stage I add a version of the manuscript with all changes marked as supplementary material.

[revised manuscript text omitted]